# Spatiotemporal patterns of water and vegetation in Poyang Lake from 2013 to 2021 using remote sensing data

Zhigang Lu[1,2,3], Zihao Chen[2,3]*, Meng Zhou[1,3], Daxing Lei[1,3], Yifan Chen[4]

1 School of Resources and Civil Engineering, Gannan University of Science and Technology, Ganzhou, Jiangxi, China, 2 School of Civil and Surveying, Jiang xi university of Science and Technology, Ganzhou, Jiangxi, China, 3 Ganzhou Key Laboratory of Remote Sensing for Resource and Environment, Ganzhou, Jiangxi, China, 4 School of Resources and Safety Engineering, Central South University, Changsha, China

* 3622140102@gnust.edu.cn

## Abstract

Continuous monitoring and research on Poyang Lake is essential to understand its ecological dynamics and promote sustainable development. Spatial and temporal dynamic monitoring and analyses of vegetation changes in the water body of Poyang Lake are still limited. This study fills this gap by using remote sensing and GIS techniques for dynamic monitoring and analysing the changes of water bodies and vegetation in Poyang Lake from 2013 to 2021. We used a combination of Maximum Likelihood Classification (MLC) and Support Vector Machine (SVM) to preprocess and classify 42 Landsat 8 OLI images. The results showed that the stability of the water body and vegetation varied greatly, with the water body showing the obvious change pattern of water rises, vegetation recedes and water recedes, vegetation grows, and the high-frequency inundation area was concentrated in the northeastern part of the lake (accounting for 60% of the total inundation area). Vegetation frequency distribution showed a pattern of sparse in the north and dense in the south, with the middle frequency area being the most, accounting for 19.88%, and the low frequency area being the least, accounting for 16.09%. The results show that the spatial and temporal distribution characteristics of water body and vegetation in Poyang Lake show low stability, which is a highly dynamic ecosystem. This study relatively makes up for the missing analysis of the stability change of water body and vegetation in the cycle of Poyang Lake, and provides a solid scientific basis for the protection and sustainable management work.

## Introduction

Wetland water resources are a crucial component of China's water supply, with freshwater lakes across the country accounting for 8% of the total freshwater storage

**Data availability statement:** All relevant data are available from https://www.gscloud.cn/ The direct links are available from the Supporting Information file titled 'links to the Image datasets.docx.

**Funding:** This research was financially supported by Jiangxi Provincial Department of Education Science and technology research Program (GJJ2403702), and Jiangxi Provincial Department of Education Science and technology research Program (GJJ2403704). The funders have no role in study design, data collection and analysis, decision to publish, or preparation of the manuscript.

**Competing interests:** The author declares no competing interests.

in the world, amounting to 22.5 billion cubic meters. Poyang Lake is one of the key water sources in the Yangtze River basin of China, playing a significant role in regulating surface and groundwater levels, maintaining soil and water balance, and reducing flood risks [1–4]. Through natural ecological filtration, it purifies water, reduces pollutants, and ensures the safety of drinking water for local communities. In addition, the Poyang Lake provides vital habitats for numerous plants and animals, serving as a crucial breeding and survival environment for many rare and endangered species, as well as being a significant habitat and stopover site for migratory birds [5–7]. Given its importance, monitoring and conserving thePoyang Lake ecosystem is essential for the sustainable development of the future ecological environment [8–12].

Satellite remote sensing technology has become a hot topic of research in monitoring lake area changes due to its ability to provide long-term, large-scale, and high-frequency data, which are essential for understanding the dynamic changes of lake ecosystems [9,13]. Poyang Lake, as the largest freshwater lake in China, has been a focal area for such studies, particularly in the context of drought and flood monitoring. For instance, Luo et al. (2022) utilized the STNLFFM model with MODIS and Sentinel-2 imagery to investigate drought conditions in Poyang Lake. Their findings revealed that drought significantly reduces the lake's water area, adversely affecting vegetation growth and the habitats of migratory birds and the Yangtze finless porpoise [14]. However, this study did not thoroughly explore the long-term ecological impacts of drought or the synergistic effects of other influencing factors, such as climate variability and human activities.

Similarly, Shankman et al. (2020) combined long-term hydrological observations and climate data to analyze the spatiotemporal trends of flood frequency in Poyang Lake and its teleconnections with global climate patterns, such as ENSO and PDO. Their research highlighted a significant increase in flood frequency over the past few decades, particularly from the late 1990s to the early 2010s, with spatial heterogeneity in flood distribution across the lake [15]. Wen et al. (2021) further analyzed the response of wetland vegetation to flood frequency changes using MODIS and Landsat data, combined with the Enhanced Vegetation Index (EVI). They identified a nonlinear relationship between EVI and flood frequency, demonstrating that moderate flooding benefits vegetation growth, while excessive flooding or prolonged inundation negatively impacts vegetation cover [16].Many studies have focused on monitoring water area changes in Poyang Lake using multi-source remote sensing data. For example, Huang et al. (2020) developed a novel method integrating active microwave and optical remote sensing data (MERSI/FY-3 and MODIS/EOS) to monitor water area dynamics [17]. Their results emphasized the advantages of remote sensing in providing large-scale, high-frequency monitoring data, but the study offered limited insights into the underlying mechanisms driving water area changes, such as climate change and human interventions.

Gu et al. (2019) analyzed water inundation frequency from 2000 to 2015 using MODIS data, focusing on spatiotemporal patterns but neglecting the integrated impacts of climate variability and human activities [18]. Wang (2021) extracted water body information from 1990 to 2020 using a decision tree-based method and

analyzed trends and driving factors for 35 lakes across China. While this study provided valuable insights into lake area changes, it lacked a detailed examination of Poyang Lake's unique ecological context [19]. Zhu et al. (2020) proposed a temperature-vegetation-water index (TVWI) for lake water extraction, which improved accuracy and stability compared to traditional methods. However, their findings were constrained by the spatial and temporal resolution of the data [20]. Wang et al. (2020) utilized synthetic aperture radar (SAR) data to monitor seasonal changes in Poyang Lake's water area, highlighting the lake's sensitivity to seasonal precipitation and Yangtze River water levels. Their study demonstrated the unique advantages of SAR data in cloud-penetrating and nighttime observations but did not fully explore the ecological implications of these changes [21].

Long-term remote sensing data have also been widely used to analyze vegetation dynamics and land cover changes. Wang et al. (2021) employed MODIS NDVI time series data to reveal long-term vegetation trends and their correlations with environmental factors, such as precipitation and temperature. Their findings provided valuable insights for land use planning and ecological conservation [22]. Aslam et al. (2023) leveraged multi-temporal remote sensing imagery and machine learning algorithms (e.g., Random Forest, Support Vector Machine) to monitor wetland dynamics. Their study demonstrated the potential of machine learning in improving the accuracy and efficiency of wetland classification and change detection, offering new methodologies for wetland monitoring and conservation [23].

Despite these advances, there are still some gaps and limitations in current research. Many studies focus on short-term changes and lack a comprehensive assessment of long-term spatial and temporal patterns and their ecological impacts, the detailed description of the intra-annual water level distribution characteristics of Poyang Lake is also not sufficiently detailed, the significant correlation relationship between quantified water, vegetation and mudflat needs to be further clarified, and there is a relative lack of information on spatial and temporal distribution characteristics of the vegetation for guiding the ecological restoration of wetlands [24]. In view of these considerations, this study will carry out related research to address the problems identified.

In this paper, the spatial and temporal patterns of water bodies and vegetation in Poyang Lake from 2013 to 2021 are investigated using high-resolution Landsat 8 images, and a classification method combining Maximum Likelihood Classification (MLC) and Support Vector Machine (SVM) is developed to improve the accuracy of the land cover mapping, to analyse the frequency distribution and stability of the water bodies and vegetation, to establish the statistical relationship, and to elucidate the interactions between them, in order to provide a scientific basis for the Poyang Lake wetland ecosystem sustainable management and protection of scientific basis.

## Study area and data

### Overview of the study area

The Poyang Lake is located in northern Jiangxi Province(28°22'～29° 45'N、115°47'～116°45'E), south of the Yangtze River (Fig 1), covering an area of 3,583 square kilometers. It is the largest lake in Jiangxi Province and one of the largest inland freshwater lakes in China. The lake connects five major rivers (Gan, Fu, Xin, Rao, and Xiu Rivers) with the Yangtze River, forming a complex and dynamic hydrological system. The surrounding area is characterized by low-lying terrain encircled by hills, with notable regional differences in landforms. The western and southern regions are dominated by river floodplains with dense coastal inlets and sedimentary silt deposits, while the eastern region features extensive marshlands with gentle slopes and a winding shoreline. The central and northern regions form the main water body of the lake, where water levels vary significantly due to seasonal hydrological changes. The climate in the Poyang Lake region is classified as subtropical humid monsoon, with distinct wet and dry seasons. The rainy season, from April to June, coincides with the hot summer months and often causes significant water level increases and flooding, while the dry season, from October to February, leads to water retreat and the exposure of mudflats. These seasonal variations create a highly dynamic environment, influencing both ecological processes and human activities in the region.

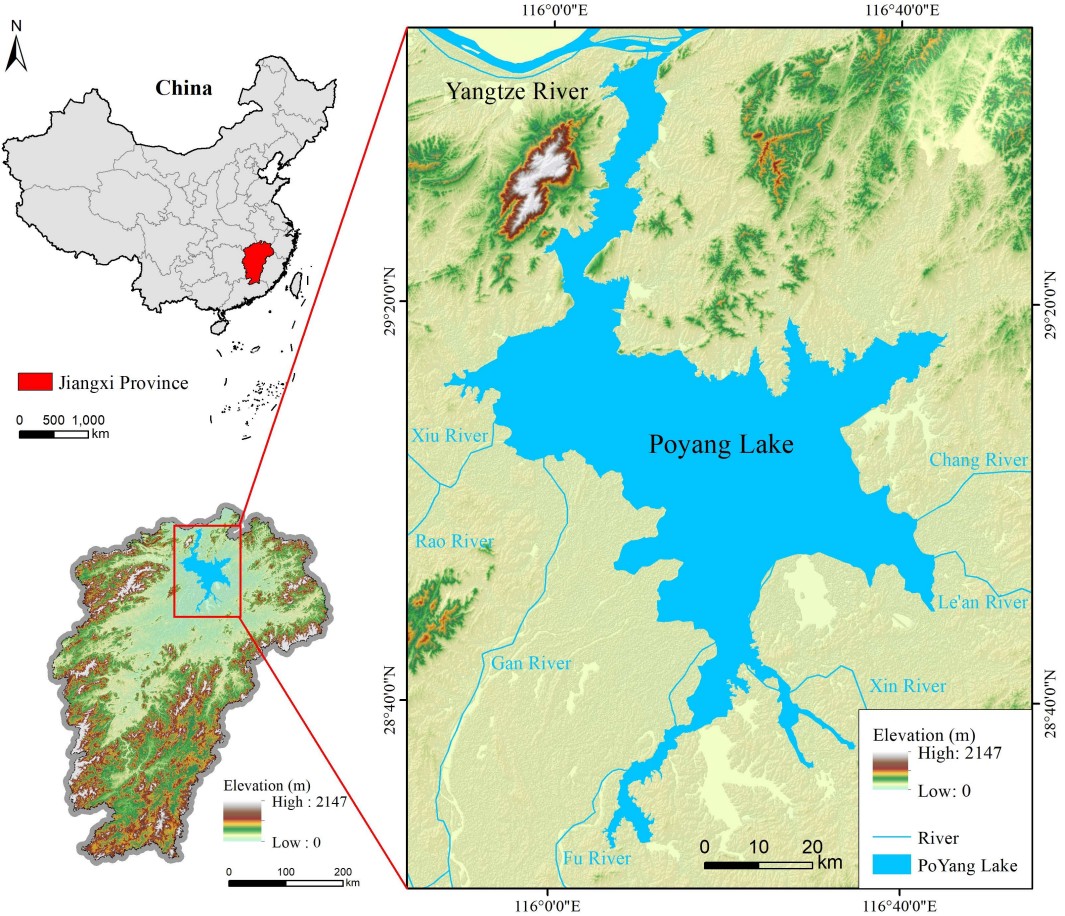

**Fig 1. Study region of Poyang Lake.**

## Data preprocessing

This study used Landsat 8 OLI remote sensing imagery as the primary data source, selecting 42 scenes covering the Poyang Lake area from 2013 to 2021. Landsat 8 OLI imagery used in this study has a spatial resolution of 30 meters for multispectral bands and 15 meters for the panchromatic band. This resolution is sufficient for capturing the spatiotemporal patterns of land cover in Poyang Lake, given the scale of the study area and the objectives of this research. The selected images were evenly distributed across each quarter of the year, with efforts made to select images from the same day and time to minimize variations in surface reflectance due to changes in solar elevation angle. Additionally, all selected images have cloud coverage below 15%. Four DEM data with GDEMV2 30M resolution of Poyang Lake area were selected to be used as the topographic reference analysis (All the image data and DEM data used for mapping are from the USGS website and are public datasets) [25]. Radiometric calibration is fundamental to the accuracy and reliability of remote sensing data, which is crucial for remote sensing applications. Using ENVI 5.3 remote sensing software, radiometric calibration was performed on the 42 Landsat 8 OLI images, followed by atmospheric correction using the FLAASH model available in ENVI 5.3, based on metadata from the images. The images were subsequently prepossessed, including clipping to the vector boundary of the Poyang Lake and band composition. Forty-two Landsat 8 OLI images of different dates were used in this study.

## Methods and classification results

### Methods flowchart

A total of 42 Landsat 8 OLI remote sensing image data were selected during 2013–2021, and after pre-processing such as radiometric calibration, atmospheric correction, band synthesis, image cropping, etc., the boundary of the lake was extracted by calculating the water body index and selecting the appropriate threshold range, and combining with the data of the artificial polder according to the results of the calculation. Supervised classification is performed on 42 images within the extracted boundary to obtain the area size change and spatial and temporal distribution of water bodies, vegetation, mudflats and other features, and the classification results are statistically and analytically analyzed in terms of inter-annual change, intra-annual change, correlation and stability. The technical method flowchart of this paper is as follow (see Fig 2).

### Classification principles

Water bodies, vegetation, and soil exhibit distinct spectral reflectance characteristics in the visible and near-infrared bands. The spectral curves of water, vegetation, and soil reveal that water bodies (especially those with low sediment content) exhibit significantly higher absorption of electromagnetic waves in the 0.4–2.5 μm wavelength range compared to most other surface features (As shown in Fig 3). Water bodies reflect very little energy in the near-infrared and mid-infrared bands, whereas vegetation and soil absorb less energy in these bands, allowing for clear differentiation between water bodies and other surface features in these spectral regions [26–29] (As shown in Figs 2 and 3).

The spectral curves of water bodies with varying sediment content (As shown in Fig 4) demonstrate that different sediment concentrations lead to variations in spectral reflectance characteristics in the visible and near-infrared bands. The Poyang Lake, being a seasonal lake, presents a unique landscape where extensive flooding and reduced water levels occur alternately in different seasons of the year. The varying area of the Poyang Lake and its sediment content across different seasons result in different colors of its water body in visible and near-infrared composite images, leading to differences in water identification criteria. For Landsat 8 imagery, the 0.64–0.67 μm range corresponds to the Red Band, while

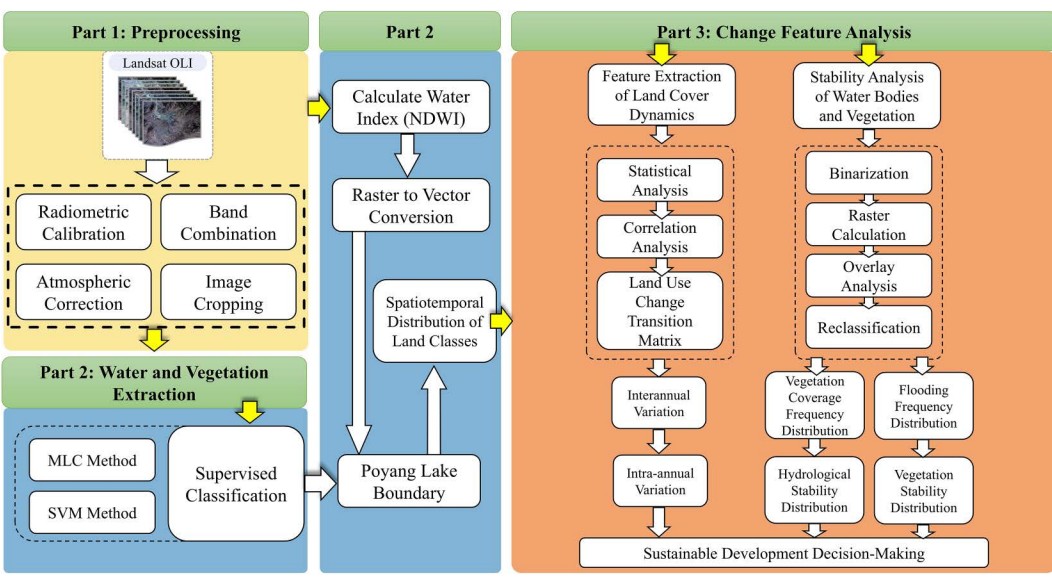

**Fig 2. Methodological framework for water-vegetation dynamics monitoring.**

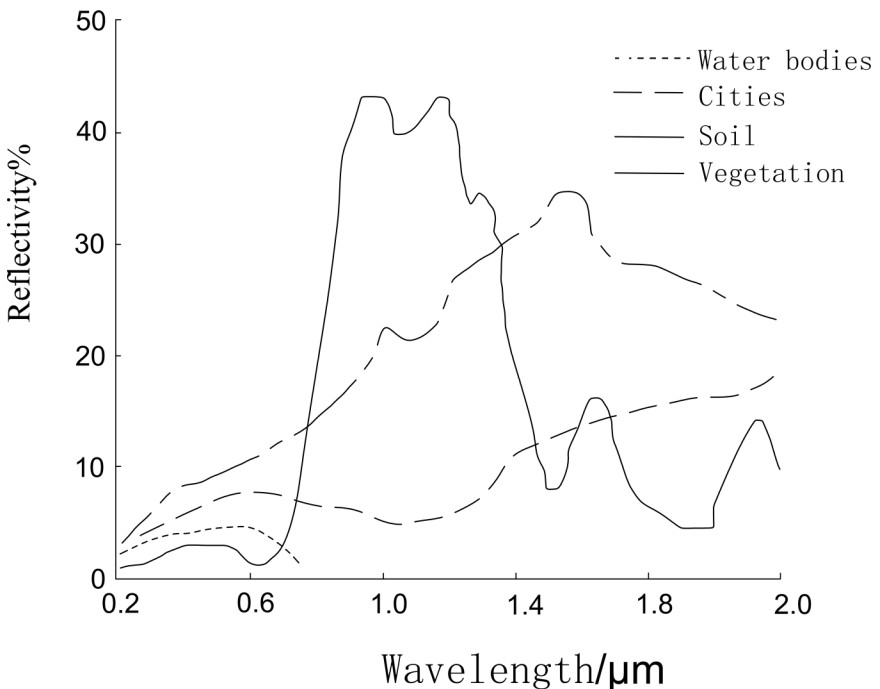

**Fig 3. Spectral reflectance curves of typical land cover types.**

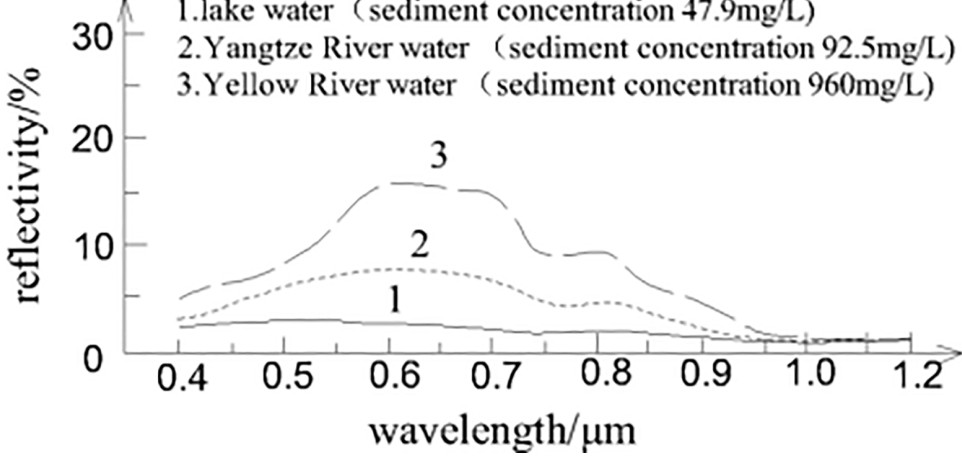

**Fig 4. Spectral reflectance curves of water bodies with different sediment concentrations.**

the 0.85–0.88 µm range corresponds to the Near Infrared Band. In the near-infrared range, water bodies exhibit significantly lower reflectance compared to vegetation, while in the red band range, water reflectance is relatively higher, and vegetation reflectance is much lower. This difference allows for effective differentiation between water bodies and vegetation.

## Maximum likelihood classification method

Let $z = \{z_i, i \in I = \{1,..., N\}\}$ be the image with selected bands mentioned before, where $i$ is the index of pixel, $I$ is the set of pixel indices, $N$ is the number of the pixels in the image, $z_i = (z_{ij}, j = 1,..., M)$ is spectral vector for the pixel $i$, $j$ is the index

of selected bands, $M$ is the number of the selected bands, $z_{ij}$ is the spectral measure of $j$'th band for pixel $i$. Let $z^S = \{z^S_l, l = 1,..., K\}$ is the set of sampling pixels, where $l$ is the index of classes, $K$ is the number of classes needed to classify, $z^S_l = \{z_i, i \in I^S_l\}$, $I^S_l$ is the index set of the sampling pixels for the class $l$.

Given the pixel $i \in \Lambda/^S$, where $I^S = \cup_{l \in \{1,...,K\}} I^S_l$, MLC classifies the pixels as [30,31],

$$L_i^{MLC} = \arg\max \{p(z_i|l), l = 1, 2, 3..., k\} \tag{1}$$

where $L^{MLC}_i$ is the class label of the pixel $i$ classified by MLC,

$$p(z_i|l) = (2\pi)^{k/2} \left| \sum_i \right|^{1/2} \exp\left[-\frac{1}{2}(z_i - u_i)^T \sum_i^{-1}(z_i - u_i)\right] \tag{2}$$

where $\mu_l$ and $\Sigma_l$ are the mean vector and covariance matrix of the samples from class $l$, respectively, that is,

$$\mu_l = \frac{1}{\#I^s_l} \sum_{i \in I^s_l} z_i \tag{3}$$

$$\sum_l = \frac{1}{\#I^s_l} \sum_{i \in I^s_l}(z_i - \mu_l)^T(z_i - \mu_l) \tag{4}$$

where # is operator for calculating the element number of a set, T is the transposition operator.

As a result, MLC classifies the image $z \backslash z^S$ as $L^{MLC} = \{L^{MLC}_i, i \in \Lambda/^S\}$.

SVM based decision tree is designed to classify the image $z \backslash z^S$, which also takes $z^S = \{z^S_l, l = 1,..., K\}$ as training set. Fig 4 shows the flowsheet of the proposed scheme.

The SVM is considered as a binary classifier (Fig 5). In this study, it is designed as non-linear SVM [32,33].

For the case that the original sample space is not linearly separable, the sample can be mapped from the original space to a higher dimensional feature space, making the sample linearly separable within this feature space. Taking $SVM_{1\text{-rest}}$ as an example, given a mapping function $\phi$, $\phi(z_i)$ can be viewed as the feature vector corresponding on $z_i$ ($i \in I^S$)

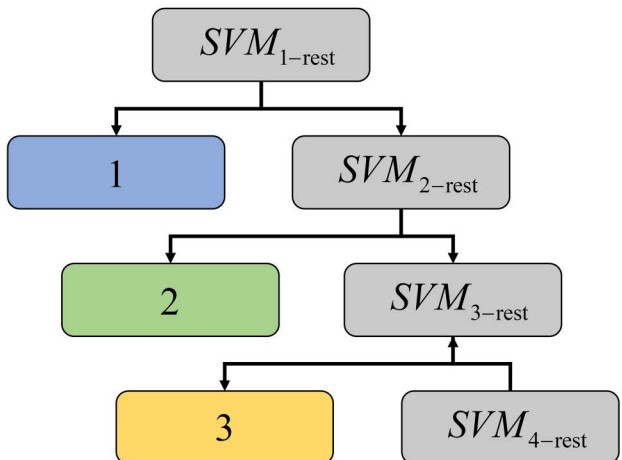

**Fig 5. The flowsheet of the SVM based decision tree.**

mapped by $\phi$. In this case, let $y_i = 1$ for $i \in I^S_1$ and $y_i = -1$ for $i \in I^S \backslash I^S_1$. Thus, The model corresponding to the dividing hyperplane in the feature space can be represented as,

$$f(z_i) = w^T \varphi(z_i) + b \tag{5}$$

where $w$ and $b$ is the parameters for the hyperplane ($w$, $b$). According to SVM theory, the learning problem of SVM can be attributed to convex quadratic programming problems, that is,

$$\min_{w,b} \frac{1}{2} \|w\|^2 \qquad s.t. y_i \left( w^T \varphi(z_i) + b \right) \geq 1, i \in I^s \tag{6}$$

The dual problem of Eq. (6) can be modeled as,

$$\max_a \sum_{i \in I^s} a_i - \frac{1}{2} \sum_{i \in I^s} \sum_{i \in I^s} a_i a_i, y_i y_i \varphi(z_i)^T \varphi(z_{i'}) \qquad s.t. y_i \left( w^T \varphi(z_i) + b \right) \geq 1, i \in I^s \tag{7}$$

Solving Eq. (7), $a$ can be obtained. Future, ($w$, $b$) can be calculated. For high-dimensional mapping function $\phi$, it is difficult to calculate $\phi(z_i)^T \phi(z_i)$. Such problems can be solved by using a kernel function $k(\cdot, \cdot)$, that is,

$$\varphi(z_{i'})^T \varphi(z_i) \leq \varphi(z_{i'}), \ \varphi(z_i) > k(z_{i'}, z_i) \tag{8}$$

By combining Eqs. (7) and (8) to solve ($w$, $b$), $f(z_i)$ for the pixels $i \in \Lambda I^S$,

$$f(z_i) = \sum_{i' \in I^s} a_{i'} y_{i'} \varphi(z_{i'})^T \varphi(z_i) + b = \sum_{i' \in I^s} a_{i'} y_{i'} k(z_{i'}, z_i) + b \tag{9}$$

Based on the $SVM_{1\text{-rest}}$, for a given $z_i$ ($i \in \Lambda I^S$), the pixels belonging to class 1 can be classified corresponding to the positive samples $\{z_i, i \in I^S_1\}$, while the pixels classified corresponding to the negative samples $\{z_i, i \in I^S \backslash I^S_1\}$ can future be classified to obtain the pixels belong to class 2 by $SVM_{2\text{-rest}}$ which is trained by sample set $\{z_i, i \in I^S \backslash I^S_1\}$, repeating the above procedure until $SVM_{(K-1)\text{-rest}}$ to classify the classes $K$-1 and $K$. Using classifiers $SVM_{1\text{-rest}}, ..., SVM_{(K-1)\text{-rest}}$ the image $z \backslash z^S$ can be classified as $L^{SVM} = \{L^{SVM}_i, i \in \Lambda I^S\}$.

Combing the classifying results from MLC and SVM, $L^{MLC} = \{L^{MLC}_i, i \in \Lambda I^S\}$ and $L^{SVM} = \{L^{SVM}_i, i \in \Lambda I^S\}$, the refined classifying results $L = \{L, i \in \Lambda I^S\}$ is constructed. That is, for the pixel $i$ with $L^{MLC}_i = L^{SVM}_i$, $L_i = L^{MLC}_i = L^{SVM}_i$, it means that the pixle i can be viewed as one correctly classified, while the pixels $\{z_i, L^{MLC}_i \neq L^{SVM}_i\}$ are reclassified with $SVM_{1\text{-rest}}, ..., SVM_{(K-1)\text{-rest}}$ which are retrained by $z^S$ and the pixels that have been correctly classified.

## Classification results

### Comparison of methods

Based on visual interpretation, Regions Of Interest (ROI) for vegetation, water bodies, mudflats, and sandbanks were established as training samples according to the landscape characteristics of the Poyang Lake area. The separability of all classification samples was calculated, ensuring that all ROI samples had good distinguishability, with separability above 1.9. During classification, two methods were used with a composite criterion, that is, each image was classified using both MLC and SVM, and the classification results were evaluated. The classification performance was evaluated based on the accuracy validation results in the Table 1, and the method with the higher accuracy was selected as the final classification method. The 42 Landsat 8 images were classified into water bodies, vegetation, mudflats, and sandbanks, resulting in classification outcomes over the nine-year period. Representative classification results for 34 of these periods are shown in Fig 6.

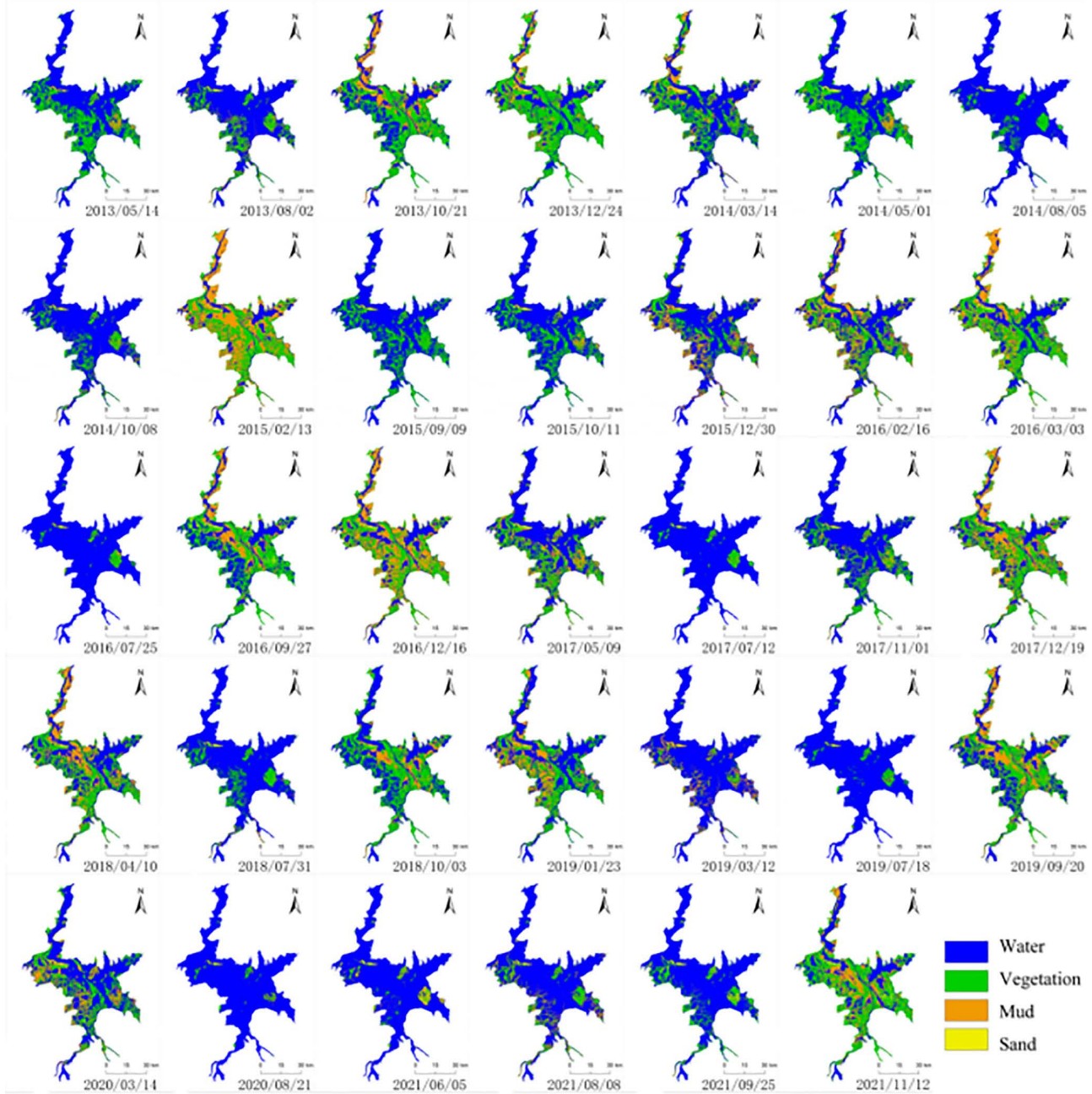

**Fig 6. Selected supervised classification results from 2013 to 2021.**

In order to evaluate the performance of land cover classification at Poyang Lake, two supervised classification methods, Maximum Likelihood Classification (MLC) and Support Vector Machine (SVM), were used. Both methods were applied to multi-temporal Landsat 8 images from 2014 to 2021, and classification accuracy was assessed using four key metrics. These metrics were derived from a confusion matrix comparing the classification results with ground truth data or high-resolution reference imagery.

As can be seen in Table 1, the classification results for the 1 May 2014 images show that the MLC method with four key accuracy metrics of 95.2%, 92.8%, 94.0%, and 0.91 outperforms the SVM method, which has key metrics of 93.5%, 91.4%, 93.2%, and 0.89. Based on these metrics, the MLC was selected as the preferred image classification for this date method. However, the results of image classification on 9 September 2015 showed that SVM nevertheless demonstrated superior performance on the four key metrics.A similar trend was observed on 10 April 2018, where SVM's accuracy (OA = 93.0%, UA = 91.8%, PA = 92.5%, and Kaapa = 0.88) was higher than that of MLC (OA = 93.0%, UA = 91.8%, PA = 92.5%, and Kaapa = 0.88), despite the fact that OA and PA of MLC were slightly higher. MLC (OA = 94.5%, UA = 93.1%, PA = 94.2%, Kaapa = 0.92).

Image classification results on 14 March 2020 showed that MLC again outperformed SVM with four key metrics of 91.8%, 89.9%, 90.2%, and 0.85, while SVM had slightly lower accuracy metrics. Finally, the image classification results on 21 August 2021 show that SVM performs best with four key metrics of 94.0%, 92.5%, 93.0%, and 0.91, while MLC has a slightly lower accuracy metric of 1–2%. Similarly, the results of the image classification on 10 April 2018 showed that although the MLC classification showed slightly higher OA and PA, the McNamara test showed that the difference was not statistically significant (p > 0.05), and therefore SVM was selected due to its higher UA and Kappa values. Finally, on 21 August 2021, SVM was selected as the preferred method due to its excellent performance on all accuracy metrics.

Based on the difference in accuracy metrics between MLC and SVM classification methods, the classification method with better accuracy metrics was selected to classify images from different time periods.MLC as a parametric method assumes that the data follows a Gaussian distribution and is very effective when this assumption holds true. In contrast, SVM, as a non-parametric method, is more robust to data with complex distributions and can handle high-dimensional feature spaces more efficiently.

This study employed the Support Vector Machine (SVM) classifier for land cover classification, given its demonstrated reliability and efficiency with small to medium-sized datasets. Accuracy assessments revealed that the SVM approach achieved over 90% overall accuracy, meeting the study's objectives. While deep learning models hold promise for further improving classification performance, their application typically requires larger labeled datasets and significant computational resources. Thus, SVM was chosen as a practical and effective method for this research. Future studies may consider incorporating deep learning models to explore their potential benefits.

## Statistical analyses

Statistical analyses of the area of water, vegetation and mudflat&sand, including mean, standard deviation, minimum and maximum values, have been carried out (see Tables 2 and 3) to characterize the distribution of the various features in Table 3.

**Table 1. Statistical results of classification accuracy comparison between MLC and SVM methods.**

| Image Date | Classification Method | OA | UA | PA | Kaapa | Chosen Method |
|---|---|---|---|---|---|---|
| 2014/05/01 | MLC | 95.2% | 92.8% | 94.0% | 0.91 | MLC |
| | SVM | 93.5% | 91.4% | 93.2% | 0.89 | |
| 2015/09/09 | MLC | 90.7% | 88.5% | 89.0% | 0.87 | SVM |
| | SVM | 92.1% | 90.3% | 91.5% | 0.89 | |
| 2018/04/10 | MLC | 94.5% | 93.1% | 94.2% | 0.92 | SVM |
| | SVM | 93.0% | 91.8% | 92.5% | 0.88 | |
| 2020/03/14 | MLC | 91.8% | 89.9% | 90.2% | 0.85 | MLC |
| | SVM | 90.2% | 88.3% | 89.5% | 0.84 | |
| 2021/08/21 | MLC | 92.3% | 91.2% | 92.1% | 0.90 | SVM |
| | SVM | 94.0% | 92.5% | 93.0% | 0.91 | |

**Table 2. Monthly Monitoring Data for Changes in area of water, vegetation and mudflat&sand between 2013–2021.**

| Image Date | Water (km²) | Vegetation(km²) | Mudflat&Sand (km²) |
|---|---|---|---|
| 2013-05-14 | 1952.7579 | 1206.6948 | 134.9064 |
| 2013-07-01 | 2758.9878 | 408.4092 | 126.9621 |
| 2013-08-02 | 2461.1868 | 562.8771 | 270.2952 |
| 2013-10-21 | 1160.5662 | 1443.4254 | 690.3675 |
| 2013-12-24 | 949.7403 | 1779.3495 | 565.2693 |
| 2014-03-14 | 1695.4803 | 1068.1722 | 530.7066 |
| 2014-05-01 | 1712.1150 | 1372.7331 | 209.5110 |
| 2014-08-05 | 2816.0667 | 397.0215 | 81.2709 |
| 2014-10-08 | 2600.2413 | 501.7707 | 192.3471 |
| 2015-02-13 | 602.5725 | 1207.3617 | 1484.4249 |
| 2015-09-09 | 2169.3159 | 1060.8678 | 64.1754 |
| 2015-10-11 | 2192.7069 | 950.7627 | 150.8895 |
| 2015-12-30 | 2053.2933 | 636.4899 | 604.5759 |
| 2016-02-16 | 1447.6608 | 902.6874 | 944.0109 |
| 2016-03-03 | 1046.7747 | 1302.9480 | 944.6364 |
| 2016-06-23 | 2944.9728 | 203.2110 | 146.1753 |
| 2016-07-25 | 3111.9939 | 138.3570 | 44.0082 |
| 2016-09-27 | 1202.9751 | 1357.1982 | 734.1858 |
| 2016-12-16 | 874.8972 | 1089.5967 | 1329.8652 |
| 2017-05-09 | 1784.1996 | 889.3161 | 620.8434 |
| 2017-07-12 | 3065.0868 | 190.6191 | 38.6532 |
| 2017-09-14 | 2308.8015 | 607.4694 | 378.0882 |
| 2017-11-01 | 2234.0142 | 742.8870 | 317.4579 |
| 2017-12-19 | 931.3137 | 1228.6692 | 1134.3762 |
| 2018-03-09 | 1351.2078 | 1048.9815 | 894.1698 |
| 2018-04-10 | 1222.9641 | 1172.0016 | 899.3934 |
| 2018-07-31 | 2651.6007 | 533.7495 | 109.0089 |
| 2018-10-03 | 1362.6405 | 1474.3026 | 457.4160 |
| 2019-01-23 | 1417.3848 | 1115.4393 | 761.5350 |
| 2019-03-12 | 2680.6977 | 204.1254 | 409.5360 |
| 2019-07-18 | 3079.5804 | 187.6563 | 27.1224 |
| 2019-08-19 | 2473.7949 | 643.9428 | 176.6214 |
| 2019-09-20 | 1132.6959 | 1223.7957 | 937.8675 |
| 2020-03-14 | 1801.0071 | 887.7033 | 605.6487 |
| 2020-04-15 | 2035.5876 | 961.5159 | 297.2556 |
| 2020-08-21 | 3163.5063 | 85.5036 | 45.3492 |
| 2020-09-06 | 3141.4338 | 98.3178 | 54.6075 |
| 2021-06-05 | 2969.0469 | 148.1166 | 177.1956 |
| 2021-08-08 | 2587.7601 | 364.8420 | 341.7570 |
| 2021-09-25 | 2661.0282 | 472.3110 | 161.0199 |
| 2021-11-12 | 1012.2894 | 1481.4981 | 800.5716 |

**Table 3. Descriptive statistics for the area of Waters,Vegetation and mudflat&Sand between 2013–2021.**

| Type | Mean (km²) | Standard deviation (km²) | Minimum (km²) | Maximum (km²) |
|---|---|---|---|---|
| Water | 2001.23 | 826.12 | 602.57 | 3163.51 |
| Vegetation | 892.34 | 476.21 | 85.50 | 1779.35 |
| Mudflat&Sand | 394.12 | 422.34 | 27.12 | 1484.42 |

The area of the water has the largest change (standard deviation of 826.12 km²), indicating that the area of the water of Poyang Lake is significantly affected by seasonal hydrological conditions. The change of vegetation area was the second largest, and the change of mudflat&sand area was the smallest, but the maximum value of mudflat&sand (1484.42 km²) was close to the minimum value of water (602.57 km²), which indicated that the mudflat sand might become the main feature type in the dry water period.

## Grubbs test

The Grubbs test is a statistical method for detecting outliers in univariate data sets and is particularly useful for identifying extreme values that may distort results. The following are the basic principles and steps of the Grubbs test [34].

1) Test statistic calculation: Calculate the sample mean $\bar{x}$ and sample standard deviation of the data set $s$, and for each data point $x_i$, calculate the standardised residuals $G_i$, and find the maximum Grubbs statistic $G_{statistic}$ from all the standardised residuals $G_i$.

$$G_i = \frac{|x_i - \bar{x}|}{S}$$

(10)

2) Critical value determination: Based on the level of significance $\alpha$ and sample size $n$, find the critical value t from the Grubbs statistic critical value table $G_{crit}$. The critical value is related to $\alpha$ and $n$.

3) Hypothesis Testing: Original hypothesis $H_0$ represents no outliers exist in the data set. Alternative hypothesis $H_1$ represents outliers exist in the data set. If $G_{statistic} > G_{crit}$, The original hypothesis was rejected as an outlier.

From Table 4, it shows that the maximum value of the Grubbs statistic for water, vegetation, mudflats and sand areas are all less than 3.04657, thus the original hypothesis cannot be rejected and it is concluded that there are no significant outliers in their data, and their values of p are also much greater than the general level of significance (e.g., 0.05), which further validates this result.

## Analysis of water and vegetation changes and stability

### Characterisation of inter-annual variability

To reflect the inter-annual variation characteristics of land cover areas in the Poyang Lake, the obtained classification results (see Fig 7) were used to calculate the number of pixels for each category by year, and the results were visualized in Fig 7.

**Table 4. Grubbs test statistics of the area of Waters,Vegetation and mudflat&Sand between 2013–2021.**

| Type | $G_{max}$ | $G_{statistic}$ | $G_{crit}$ | P |
|---|---|---|---|---|
| Water | 602.5725 | 1.8408 | 3.04657 | 2.51952 |
| vegetation | 1779.3495 | 2.06952 | 3.04657 | 1.40878 |
| Mudflat&sand | 1484.4249 | 2.65233 | 3.04657 | 0.23209 |

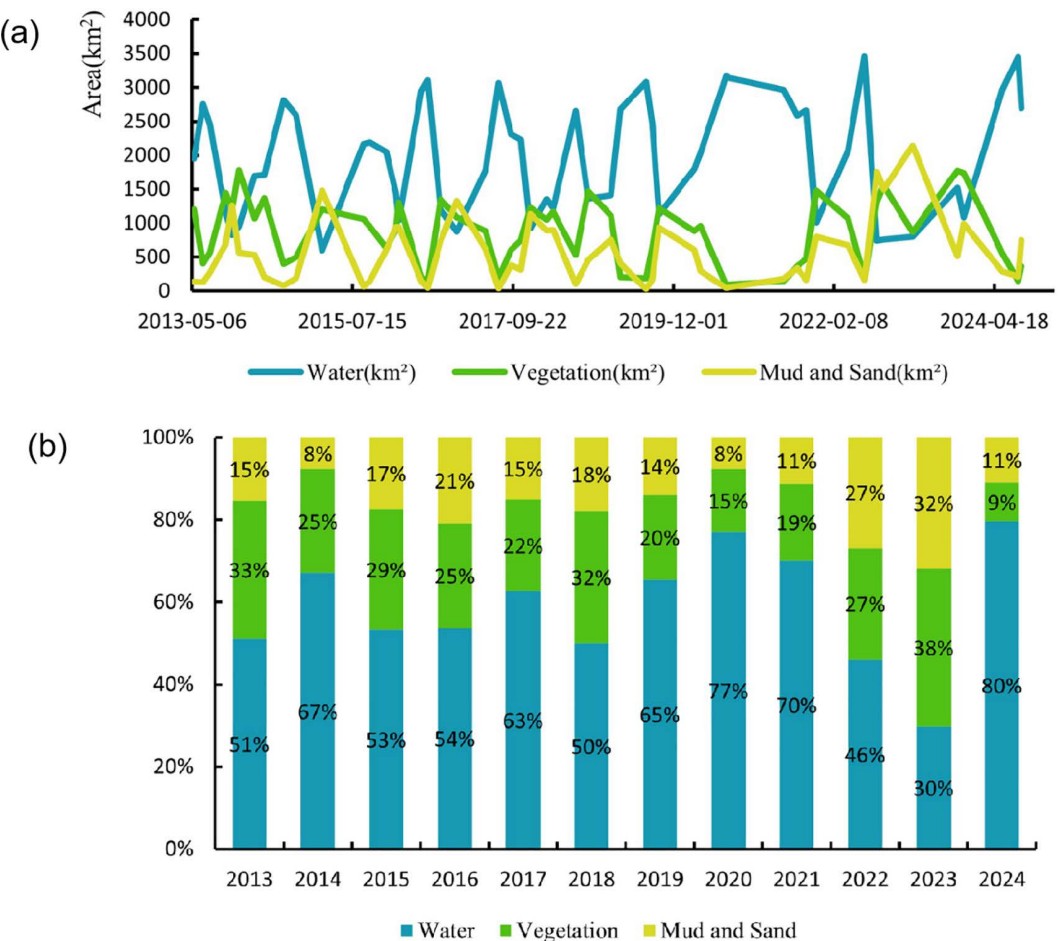

**Fig 7. The inter-annual variation characteristics of land cover areas changes.** (a) Statistical chart of inter-annual area changes.(b)Statistical chart of inter-annual area share.

Several characteristics of the area distribution of the regional category of Poyang Lake can be summarized from the curve changes(see Fig 7a). The area curve of the water body shows significant periodicity, and the area curve of vegetation and sediment changes with the periodic change of the water body. the area curve of the water body and the area curve of vegetation have more obvious phenomena of water rising and grass receding, and water receding and grass growing. When the area curve of the water body rises, the corresponding curve of vegetation and sediment decreases. The area curves of vegetation and sediment show intermittent similarity, and the size of the area is close to each other.

From 2013 to 2021, the percentage of watershed area in the Poyang Lake area fluctuates greatly, showing relative instability, while the percentage of vegetation and mudflat is relatively stable with less fluctuation (see Fig 7b). Specifically, the average annual percentage of watershed area fluctuated between 50% and 77%, with a maximum fluctuation of 27%. The mean annual percentage of vegetated area varied between 15 and 33 per cent, with a maximum fluctuation of 18 per cent. The average annual percentage of mudflats and sandbars, on the other hand, fluctuated between 8 and 21 per cent, with a maximum fluctuation of 13 per cent.

In 2014, 2019, 2020, and 2021, watersheds are more abundant, with average annual percentages exceeding 65%. Especially in 2020, the watershed area reached 77%, the highest value during the study period. This indicates that water

resources in the Poyang Lake area are relatively abundant in these years, which may be related to the increase in precipitation or water management measures in that year.

The fluctuation of vegetation area was relatively small, but showed a more significant decrease in 2016 and 2018, down to 15% and 16%, respectively. This may be related to climatic conditions or human activities (e.g., agricultural expansion) in that year. The area of mudflats and sandbars increased in 2015 and 2017 to 21% and 20%, respectively, which may be related to declining water levels or increased sedimentation.

The results show that the fluctuations in watershed area reflect the instability of hydrological conditions in the Poyang Lake area, while the relative stability of vegetation and mudflats suggests that these ecosystems have been able to adapt to changes in water levels to some extent. However, the significant fluctuations in water area may pose challenges to local ecosystem and water resource management, and further research and monitoring are needed to develop effective management strategies.

## Correlation analysis

In order to analyse these changes in more detail, the standard deviations for each category have been calculated. The standard deviation is 9.5 per cent for watershed area, 6.2 per cent for vegetated area and 4.8 per cent for mudflats and sandbars. These statistical indicators further confirm that the instability of watershed area is higher than that of vegetation and mudflats. In addition, correlation analyses of watershed area with precipitation and water management policies may help to understand the causes of its fluctuations.

In the Fig 8a, there was a significant negative correlation between the area of water and the area of vegetation (pearson's r=-0.921, $R^2$=0.84), indicating that as the area of water increased, the area of vegetation decreased and vice versa. There was also a negative correlation (pearson's r=-0.881, $R^2$=0.77) between the area of water and the area of mudflat, indicating that the area of mudflat and sand decreased when the area of water increased (see Fig 8b).The area of vegetation was positively correlated with the area of mudflat and sand (Pearson's r=0.628, $R^2$=0.40), suggesting that vegetation and mudflat and sand may co-occur under certain circumstances (e.g., during dry periods) (see Fig 8c).

The results show that during the abundant water period, the water increases in area, and the elevated water surface submerges the mudflats and vegetation, resulting in a decrease in the area of mudflat and sand and vegetation; during the dry water period, the water level decreases, exposing the mudflat and sand and vegetation. So significant negative correlation is showed between water and mudflat and sand, and water and vegetation. For the vegetation and mudflat, during the abundant water period, both vegetation and sediment are submerged, and during the dry water period, the mudflat and sand is exposed after the water recedes, while the growth of vegetation has a time lag, and it usually takes some time after the water recedes before it starts to grow, and the growth rate and distribution of different kinds of plants are different, so there is not a simple linear relationship between the vegetation and the sediment, which leads to the linear distribution of the vegetation and mudflat and sand is not significant.

## Matrix of land category transfers

Taking 2017 as an intermediate point, the changes in land categories between the two five-year periods of the first five years (2013–2017) and the second five years (2017–2021) are studied. Based on the inter-annual average area of each land category derived from the Table 3, with the help of land use transfer matrix, the land category shifts between the two five-year periods are given in the Tables 5 and 6:

During the period 2017–2021, vegetation reversed its previous decline and increased significantly, with a decrease in losses. This indicates improved environmental conditions or effective conservation measures. Mudflat dynamics remained high in both periods, but there was a significant increase in mudflat area during 2017–2021, which may be due to sedimentation or a decrease in water levels.There was a significant increase in the area of sandy areas during the period 2017–2021, which contrasts with the stability of the previous period. It indicates a change in sediment transport or

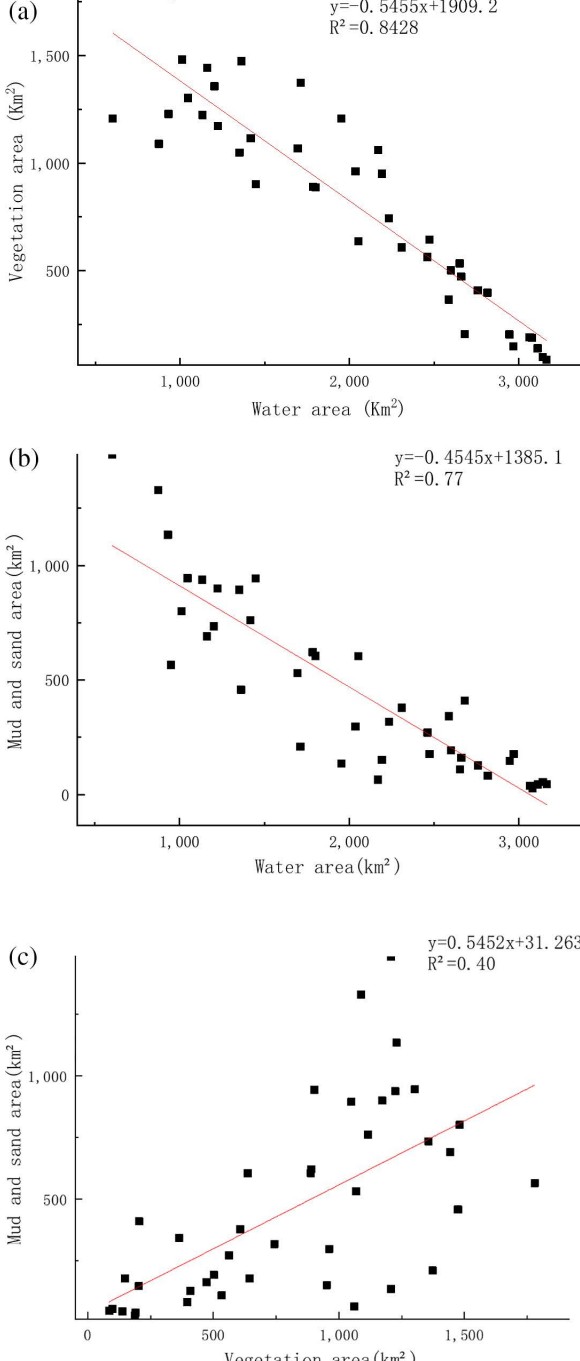

**Fig 8. Intercorrelation between the areas of water, vegetation and sediment.** (a) Correlation between vegetation and water. (b) Correlation between Mud&sand and water. (c) Correlation between vegetation and Mud&san.

erosion pattern. The expansion of water bodies observed during the period 2013–2017 was reversed during the period 2017–2021 with a significant decrease in water, a phenomenon that reflects changes in hydrological conditions or human intervention.

**Table 5. Matrix of land category transfers between 2013–2017.**

| 2013-2017 | Vegetation(km²) | Mudflat(km²) | Sand(km²) | Water(km²) | Total (km²) | Decreasing (km²) |
|---|---|---|---|---|---|---|
| vegetation | 483.20 | 140.91 | 0.17 | 585.37 | 1209.65 | 726.45 |
| Mudflat | 250.13 | 143.50 | 2.86 | 844.01 | 1240.50 | 1097.00 |
| Sand | 1.13 | 4.19 | 8.68 | 6.49 | 20.48 | 11.80 |
| Water | 8.43 | 16.92 | 0.23 | 798.15 | 823.73 | 25.58 |
| Total | 742.89 | 305.52 | 11.94 | 2234.01 | 3294.36 | |
| Increasing | 259.69 | 162.02 | 3.26 | 1435.86 | | |

**Table 6. Matrix of land category transfers between 2017–2021.**

| 2017-2021 | Vegetation(km²) | Mudflat(km²) | Sand(km²) | Water (km²) | Total(km²) | Decreasing(km²) |
|---|---|---|---|---|---|---|
| vegetation | 700.16 | 19.20 | 16.68 | 6.86 | 742.89 | 42.73 |
| Mudflat | 223.23 | 35.14 | 35.53 | 11.62 | 305.52 | 270.38 |
| Sand | 0.58 | 0.27 | 10.59 | 0.50 | 11.94 | 1.35 |
| Water | 557.54 | 644.33 | 38.83 | 993.31 | 2234.01 | 1240.70 |
| Total | 1481.50 | 698.95 | 101.62 | 1012.29 | 3294.36 | |
| Increasing | 781.34 | 663.81 | 91.03 | 18.98 | | |

The data results show the dynamic nature of land use changes, particularly in the mudflat and watershed categories. These changes may be influenced by natural processes (e.g., sedimentation, flooding) and human activities (e.g., land reclamation, water management).The recovery of vegetation during the period 2017-2021 is a positive trend, indicating the effectiveness of conservation efforts or natural regeneration. However, continuous monitoring is necessary to ensure sustained recovery. Expansion of sandy areas and dynamic fluctuations in mudflats require further investigation into the drivers behind them, such as climate change, coastal erosion or sediment management practices. Policy makers and land managers should take these trends into account when planning for sustainable land use and environmental protection, especially in areas prone to dynamic changes.

## Characterisation of intra-year changes

To reflect the intra-annual variation characteristics of land cover areas in the Poyang Lake, the classification results were used to calculate the number of pixels for each category by month, and the results were visualized (see Fig 9).

The areas of water and mudflat-sandbar display pronounced seasonal variability, and the areas of vegetation exhibit relatively stable temporal dynamics rule. The proportion of water areas increased from 23% in January to 66% in December, with the lowest value observed in January (23%) and the highest value in December (66%). This pattern suggests that water areas are relatively smaller in early winter and spring but significantly larger in summer and autumn. The standard deviation of monthly water area changes was 15.2%, indicating substantial monthly fluctuations. The marked seasonal variation in water areas is likely associated with seasonal changes in precipitation and water resource management policies. Increased rainfall during summer and autumn may contribute to the expansion of water areas.

The proportion of vegetation areas fluctuated between 37% and 55% from January to December, with the highest values observed in May and December (55%) and the lowest value in January (37%). This indicates that vegetation cover remains relatively stable throughout the year, with slight increases in spring and winter. The standard deviation of monthly vegetation area changes was 6.8%, suggesting relatively minor monthly variations. The relative stability of vegetation

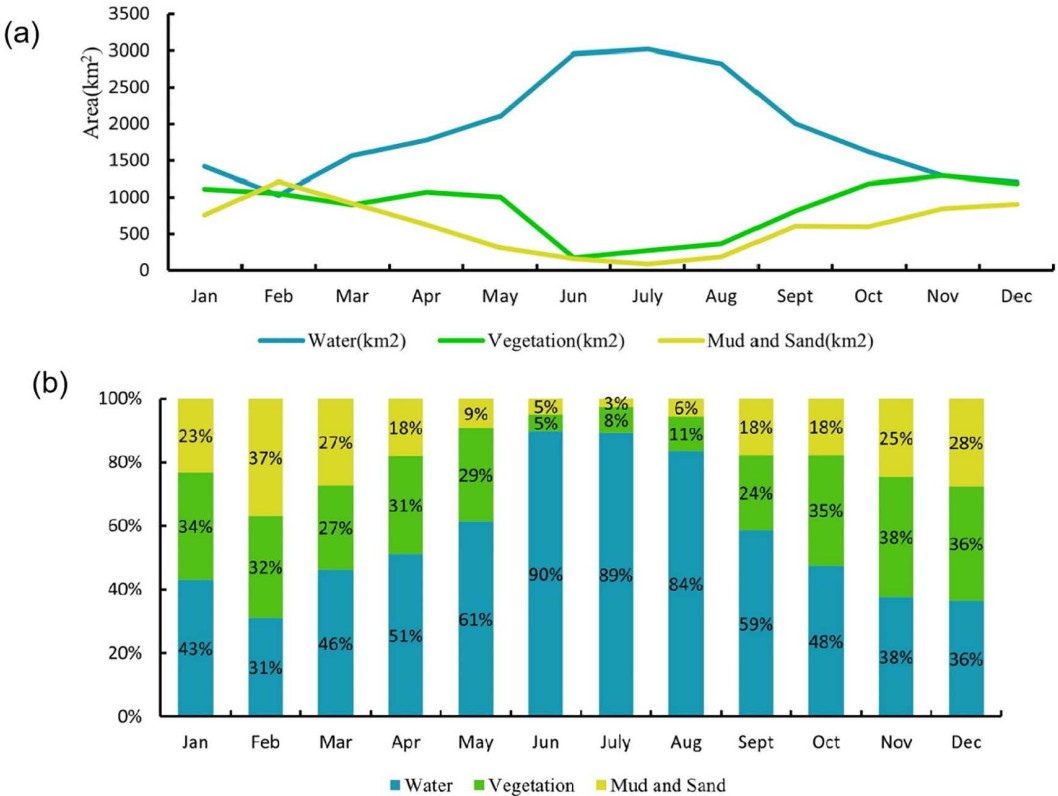

**Fig 9. The intra-annual variation characteristics of land cover areas.** (a) Statistical chart of changes in area during the year. (b) Statistical chart of area share during the year.

areas may reflect the resilience of vegetation to water level changes, while the slight increases in spring and winter could be related to the growing season and climatic conditions.

The proportion of mudflat-sandbar areas decreased from 10% in January to 13% in December, with the highest value in January (10%) and the lowest value in December (13%). This suggests that mudflat-sandbar areas are larger in winter and smaller in summer and autumn. The standard deviation of monthly mudflat-sandbar area changes was 4.5%, indicating relatively minor monthly fluctuations. The seasonal variation in mudflat-sandbar areas may be related to water level changes and sediment transport dynamics, with larger areas exposed during lower water levels in winter.

The year-round patterns of change in water, vegetation and mudflat-sandbar areas all show distinct seasonal characteristics, which has been influenced by a combination of various environmental and man-made factors such as climatic and human activities.

## Frequency analysis of water bodies and vegetation

The classification results from the 42 images were binarized, assigning a value of 1 to water pixels and 0 to non-water pixels. Subsequently, the annual mean for each pixel was calculated. The binary results for each year were aggregated into a raster, and the values were accumulated across nine years to calculate new values for each pixel, followed by reclassification of the results [35]. The new values ranged from [0, 9], representing the frequency of occurrence of either water or vegetation for each pixel, with higher values indicating more frequent classification as vegetation, thus implying a higher likelihood of the pixel being either water or vegetation during the nine-year period. A value of 0 indicates no distribution of

water or vegetation during the nine years, and thus, it was set as null. For water bodies, frequencies of (0, 3], (3, 5], (5, 7], and (7, 9] correspond to low, medium, relatively high, and high-frequency distributions, respectively. For vegetation, frequencies of (0, 3], (3, 5], (5, 7], and (7, 9] correspond to low, relatively low, medium, and high-frequency distributions, respectively. The resulting frequency distribution maps for water and vegetation are shown in Fig 10.

For water bodies, in terms of temporal distribution, the water coverage was abundant between 2019 and 2021, whereas the overall water distribution was less extensive from 2013 to 2018. Spatially, the frequency of water distribution exhibited a "northeast high, southwest low" pattern, with relatively high and high-frequency inundation areas mainly concentrated in the northeastern part of the Poyang Lake area, while scattered in other regions. Low and medium-frequency inundation areas were primarily distributed in the southwest and southeast, showing a network structure. In terms of quantity, the inundated area of the Poyang Lake was extensive, with an inundation rate of over 97% within the study area. Pixels with relatively high and high-frequency inundation (indicating a water occurrence probability greater than 55%) accounted for nearly 60%, while low or medium-frequency inundation pixels accounted for about 37%. For vegetation, temporally, vegetation was dense between 2013 and 2018, whereas it was relatively sparse from 2019 to 2021. Spatially, the vegetation frequency distribution showed a "sparse north, dense south" pattern, with medium and high-frequency areas mainly concentrated along riverbanks, eastern marshes, and exhibiting a network structure around enclosed ponds. In terms of quantity, the proportion of pixels for each frequency class ranged from 16% to 20%, with medium-frequency areas accounting for the highest proportion at 19.88%, and relatively low-frequency areas having the smallest proportion at 16.09%, with the overall proportions across different frequencies being relatively similar.

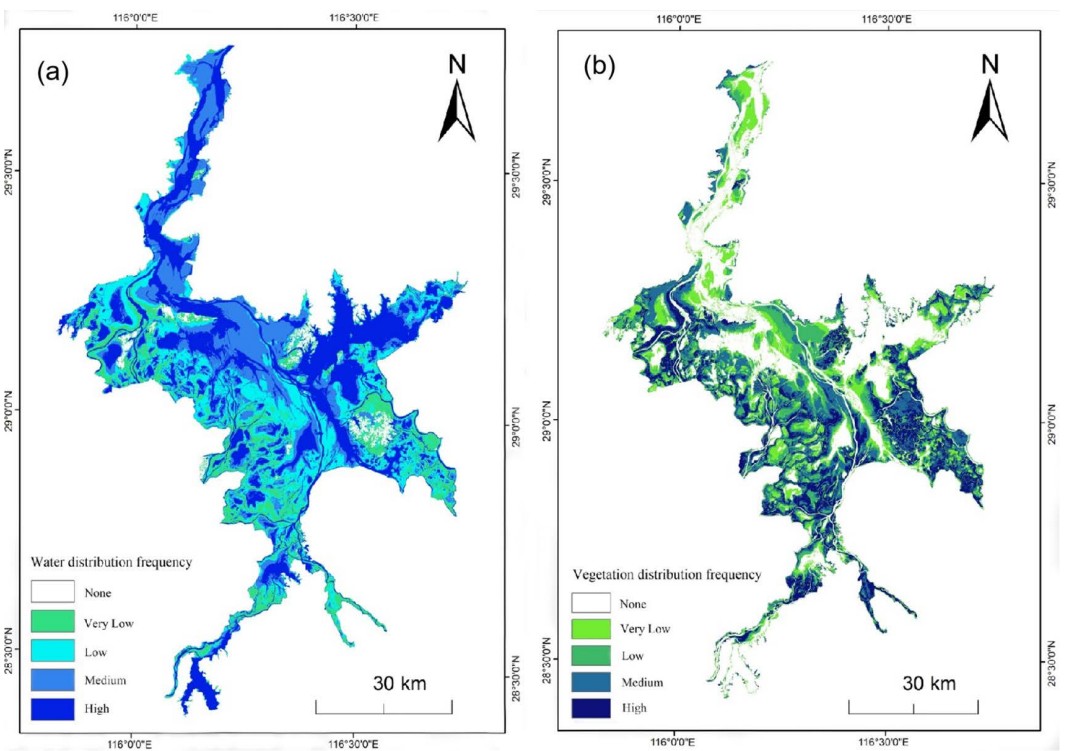

**Fig 10. Frequency Distribution of Water Bodies and Vegetation from 2013 to 2021.** (a) Frequency distribution of water bodies and vegetation in 2013.(b) Frequency distribution of water bodies and vegetation in 2021.

## Stability analysis of water bodies and vegetation

Coefficient of Variation (*CV*) reflects the degree of dispersion and relative fluctuation in data distribution. In this study, the *CV* was used to analyze the stability of land cover types in the Poyang Lake on a per-pixel basis. The calculation formula is as follows,

$$CV = \frac{1}{\overline{F}}\sqrt{\frac{\sum_{i=1}^{n}\left(F_i - \overline{F}\right)^2}{n}} \times 100\%$$

(9)

where *CV* represents the coefficient of variation for land cover types, *n* represents the number of years (in this study, $n = 9$), *F* represents the average annual land cover variation frequency over *n* years, and *Fi* represents the coverage frequency for year *i*. A smaller *CV* indicates less dispersion and greater stability in the data, whereas a larger *CV* indicates greater dispersion and lower stability [36].

To explore the stability distribution of water bodies, *CV* was calculated on a per-pixel basis to determine the variability of water bodies, resulting in the stability distribution of water. Similarly, the variability of vegetation was calculated on a per-pixel basis, resulting in the stability distribution of vegetation, with the number of pixels for each category (see Fig 11).

Based on the spatial characteristics of the Poyang Lake water bodies, it can be concluded that the northwestern, northeastern, and some southern regions exhibit high water stability, indicating that these areas have relatively stable water quality, which is beneficial for ecological conservation and the sustainable use of water resources. In contrast, the central mudflat areas are primarily characterized by low or relatively low water stability, possibly influenced by human activities and geomorphology, requiring focused attention and management to prevent water quality degradation and damage to the ecosystem. Overall, the water bodies in the Poyang Lake show uneven spatial distribution, underscoring the need for differentiated management of water environments in different regions [37]. In terms of quantitative characteristics, the proportion of the Poyang Lake water bodies with high stability is 24.91%, those with medium stability account for 35.60%, and those with low or relatively low stability account for 39.49%. With over 70% of the water bodies falling in medium or low stability regions, the overall stability of the Poyang Lake water is relatively low, making it a highly dynamic lake.

Based on the spatial characteristics of vegetation in the Poyang Lake region, it can be concluded that the central-southern and eastern marsh areas of the lake are primarily covered by vegetation with relatively high and high stability. These areas have good vegetation coverage, indicating favorable environmental conditions that support plant growth and ecosystem stability. In contrast, the northern river inlets, central mudflats, and along tributaries are mainly characterized by low or relatively low stability vegetation, possibly affected by hydrodynamic forces and negative human disturbances. These areas require enhanced ecological restoration and protection efforts to promote healthy vegetation growth and restore ecological balance. Overall, vegetation in the Poyang Lake region shows uneven spatial distribution, emphasizing the need for targeted management and protection of vegetation in different areas [38]. In terms of quantitative characteristics, the proportion of vegetation with relatively high stability is 15%, medium stability is 17.51%, and low or relatively low stability is 38.86% [39]. The proportion of pixels with medium to low stability is significantly higher than those with relatively high and high stability. Therefore, the overall stability of vegetation in the Poyang Lake is also relatively low, and the vegetation null value areas overlap considerably with high stability water areas, suggesting a significant correlation between the stability of vegetation and water [40].

## Discussion

This study systematically analyses the spatial and temporal distribution characteristics of the water body and vegetation cover of Poyang Lake from 2013 to 2021 based on 42-phase Landsat 8 OLI images, using the MLC and SVM supervised classification methods. The results are consistent with the existing literature, for example, [Zhang et al., 2020] pointed out that the area of water body in Poyang Lake is significantly negatively correlated with the vegetation cover, and the present

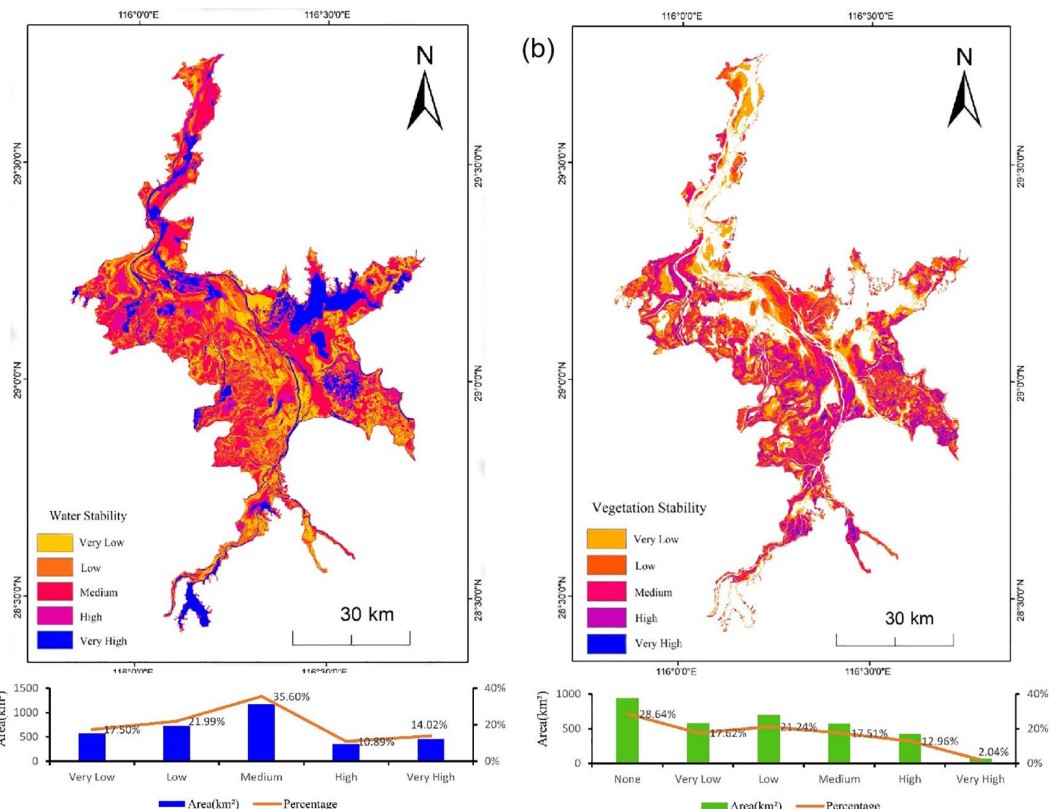

**Fig 11. Stability distribution statistics of water bodies and vegetation from 2013 to 2021.** (a) Statistics on the distribution of water bodies stability from 2013 to 2021. (b) Statistics on the distribution of vegetation stability from 2013 to 2021.

study further quantifies this relationship, and reveals the dynamic patterns of low vegetation in rising water and high vegetation in receding water [41]. In addition, [Li et al., 2019] found that the seasonal variation of water bodies in Poyang Lake was significant, and this study clarified the specific distribution characteristics of the high-water period (June-September) and low-water period (November-March) through the pixel-level frequency calculations, which supplemented the insufficient details of the existing studies on the intra-annual variation of Poyang Lake [42].

In terms of spatial distribution, the frequency distribution pattern of high in the northeast and low in the southwest found in this study is consistent with the results of [Wang et al., 2021], but this study further reveals the lower stability of the water body, indicating that Poyang Lake is a highly dynamic lake system [43]. The distribution pattern of sparse in the north and dense in the south of vegetation is consistent with that of [Chen et al., 2018], but the stability analysis in this study clarifies that the high stability vegetation is mainly distributed in the south-central and eastern marsh areas, which provides a more accurate spatial guidance for wetland ecological restoration [44].

The results of this study are of great practical significance to the management of wetlands in Poyang Lake, and the negative correlation between the water and vegetation revealed by this study provides a scientific basis for the regulation of ecological water level. For example, the area of water can be increased by artificial regulation in low water level period to promote the recovery of wetland vegetation, while the water level needs to be controlled in high water level period to avoid excessive inundation of vegetation. Based on the stability distribution of water bodies and vegetation, ecological restoration projects can be prioritised in low-stability areas (e.g., the northern river inlet and the central mudflat) to improve the overall stability of the wetland ecosystem. The inter-annual change characteristics revealed by the results show that

Poyang Lake is highly sensitive to climate change. In the future, management strategies should incorporate climate change adaptation measures, such as the establishment of a dynamic monitoring system, timely adjustment of water level regulation and vegetation restoration programmes.

This study has several limitations that should be addressed in future research. First, the analysis relied solely on Landsat 8 imagery, which, while offering consistent temporal resolution, has a spatial resolution of 30 meters that may not capture fine-scale land cover changes. Incorporating higher-resolution imagery, such as Sentinel-2, could provide more detailed spatial insights. Second, the classification approach used (MLC and SVM) provided reliable results but did not leverage advanced techniques such as deep learning, which could further improve classification accuracy, particularly in complex or mixed land cover areas. Finally, this study did not account for external drivers of land cover changes, such as climate variability or human activities, which should be incorporated into future analyses to better understand the underlying mechanisms of spatiotemporal dynamics.

## Conclusion

This study, based on 42 periods of Landsat 8 OLI imagery, investigated the distribution of water bodies and vegetation in the Poyang Lake, China's largest freshwater lake, from 2013 to 2021. Two supervised classification methods, MLC and SVM, were used for comparison and validation to derive the spatiotemporal distribution of land cover types in the Poyang Lake over nine years. Subsequently, statistical techniques and analytical methods were employed to analyze the inter-annual and intra-annual variations in the Poyang Lake region, and pixel-wise calculations were conducted to determine the frequency and variability of water and vegetation, leading to a concise analysis of their frequency and stability distributions. The results from 2013 to 2021 indicate as follows.

1) The water bodies in the Poyang Lake region showed significant inter-annual periodicity, with a strong negative correlation between water area and vegetation/mudflat areas, characterized by the "water rising, vegetation retreating" and "water receding, vegetation growing" patterns.

2) Interannually, water body variations were substantial, with a maximum fluctuation of 27%. The water body was abundant in 2014, 2019, 2020, and 2021, while it shrank in 2013, 2015, 2016, 2017, and 2018. Intra-annually, the period from June to September was the high-water season, and from November to March was the low-water season. Vegetation coverage was minimal during the high-water season, almost entirely submerged, while outside the high-water season, it remained stable at around 30%.

3) The spatial frequency distribution of water bodies in the Poyang Lake exhibited a "northeast high, southwest low" pattern, with most high and relatively high-frequency inundation areas concentrated in the northeastern part of the Poyang Lake. Low to medium-frequency inundation areas were concentrated in the southwest and southeast, forming a network structure along river courses. Vegetation frequency distribution showed a "sparse north, dense south" pattern, with medium and high-frequency vegetation areas mainly along riverbanks and in eastern marshes, forming a network structure around enclosed ponds.

4) The overall stability of the Poyang Lake water bodies was relatively low, indicating a highly dynamic lake, with more than 70% of the area falling into medium or low stability categories. High stability water bodies were mainly located in the northwestern, northeastern, and small parts of the southern region, while low and relatively low stability water bodies were primarily distributed in central mudflat areas. Similarly, the stability of vegetation was also low, with medium and low stability regions far exceeding relatively high and high stability areas. Vegetation with relatively high and high stability was mainly distributed in the central-southern and eastern marsh areas of the lake, while low and relatively low stability vegetation was mainly found at northern river inlets, central mudflats, and along tributary banks.

The spatiotemporal distribution of water bodies and vegetation in the Poyang Lake region, as revealed by this study, offers valuable insights for sustainable development and management of the area. It provides a scientific foundation for wetland conservation and restoration, playing a crucial role in expanding wetland areas, restoring wetland ecosystems, and maintaining the ecological balance of the Poyang Lake.

The findings of this study provide a comprehensive understanding of the spatiotemporal dynamics of water bodies and vegetation in Poyang Lake. Compared to existing literature, this research offers detailed insights into the specific patterns and drivers of wetland dynamics in this region. The results can inform adaptive management strategies and long-term monitoring networks for the Poyang Lake [45]. Future research should focus on integrating multi-source data to enhance classification accuracy and exploring advanced machine learning models to improve the identification of complex wetland types. Additionally, quantifying the contributions of climate change and human activities to wetland dynamics and assessing ecosystem services will further support sustainable management efforts.

## Supporting information

**S1 Data.**
(DOCX)

## Acknowledgments

We thank all members of the Multimodal Remote Sensing Data Fusion for Green Mining and Efficient Resource Utilisation in Mines team and two anonymous reviewers for their valuable comments on the previous version of this manuscript.

## Author contributions

**Conceptualization:** Zhigang Lu.

**Data curation:** Zihao Chen.

**Formal analysis:** Daxing Lei.

**Methodology:** Meng Zhou.

**Writing – original draft:** Zihao Chen.

**Writing – review & editing:** Yifan Chen.

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
