## [Decision Letter · Decision Letter 0]

PONE-D-24-53468Spatiotemporal Patterns of Water and Vegetation in Poyang Lake Using Remote Sensing DataPLOS ONE

Dear Dr. lu,

Thank you for submitting your manuscript to PLOS ONE. After careful consideration, we feel that it has merit but does not fully meet PLOS ONE’s publication criteria as it currently stands. Therefore, we invite you to submit a revised version of the manuscript that addresses the points raised during the review process.

We look forward to receiving your revised manuscript.

Kind regards,

Yizhi Sheng

Academic Editor

PLOS ONE

Journal Requirements:

“This paper gets its funding from Jiangxi Provincial Department of Education Science and technology research Program (GJJ2403702); Jiangxi Province Higher Education Teaching Reform Research Project (JXJG-22-36-4). The authors wish to acknowledge these supports”

5. We note that Figures 1, 4, 7 and 8 in your submission contain map images which may be copyrighted. All PLOS content is published under the Creative Commons Attribution License (CC BY 4.0), which means that the manuscript, images, and Supporting Information files will be freely available online, and any third party is permitted to access, download, copy, distribute, and use these materials in any way, even commercially, with proper attribution. For these reasons, we cannot publish previously copyrighted maps or satellite images created using proprietary data, such as Google software (Google Maps, Street View, and Earth). For more information, see our copyright guidelines: http://journals.plos.org/plosone/s/licenses-and-copyright .

    a. You may seek permission from the original copyright holder of Figures 1, 4, 7 and 8 to publish the content specifically under the CC BY 4.0 license. 

We recommend that you contact the original copyright holder with the Content Permission Form (http://journals.plos.org/plosone/s/file?id=7c09/content-permission-form.pdf ) and the following text:

Reviewers' comments:

Reviewer's Responses to Questions

**Comments to the Author**

1. Is the manuscript technically sound, and do the data support the conclusions?

Reviewer #1: Yes

Reviewer #2: Partly

2. Has the statistical analysis been performed appropriately and rigorously? 

Reviewer #1: Yes

Reviewer #2: No

3. Have the authors made all data underlying the findings in their manuscript fully available?

Reviewer #1: No

Reviewer #2: Yes

4. Is the manuscript presented in an intelligible fashion and written in standard English?

Reviewer #1: Yes

Reviewer #2: Yes

5. Review Comments to the Author

Reviewer #1: This paper investigated the distribution of water bodies and vegetation in the Poyang Lake from 2013 to 2024. The comments for the authors’ reference are listed below.

(1). The spatial distributions of wetland cover types were analyzed for 2013 to 2021. It’s suggested that the years 2021 to 2024 be included as well, as the recent patterns are of firm importance and practical meaning for understanding the Poyang Lake region.

(2). The authors adopted the maximum likelihood classification and SVR classification to derive the spatial distribution of wetland cover types. Why not use the state-of-the-art deep learning models whose strong classification performances have been well demonstrated?

(3). A paragraph briefly showing the structure of the paper could be added at the end of the Introduction section.

(4). The spatial resolution of the Landsat 8 OLI data should be given in the Data Preprocessing section.

(5). There are many LULC produces publicly available. Some of them are with high spatial resolutions. Why not use these products to investigate the spatiotemporal patterns?

(6). Limitations of the research should be indicated in the Conclusions section.

Reviewer #2: Title and Abstract:

The title could be more specific - consider adding the study period (2013-2021)

Abstract needs better organization with clearer methods and results sections

Quantitative results in the abstract should include specific values/percentages

Introduction (Lines 1-105):

The literature review section needs better organization and flow

More recent citations needed (many are from pre-2020)

Research gaps and objectives need to be stated more explicitly

Hypothesis or research questions should be added

Better transition needed between background and study objectives

Please follow the literature review by a clear and concise state of the art analysis. Clearly discuss what the previous studies that you are referring to. Recent research works need to be cited. There are few manuscripts suggested for this research:

https://doi.org/10.1016/j.jenvman.2024.123123

https://doi.org/10.1007/s11356-023-27554-5

https://doi.org/10.3390/rs16050928

https://doi.org/10.1016/j.gr.2023.12.015

https://doi.org/10.3390/w16010063

https://doi.org/10.1080/19475683.2023.2166989

https://doi.org/10.1016/j.bdr.2023.100416

https://doi.org/10.1016/j.rama.2024.02.008

https://doi.org/10.1007/s43832-024-00110-z

https://doi.org/10.1007/s10708-024-11203-6

Methods (Lines 106-250):

Study area description needs more specific geographic coordinates and characteristics

Data preprocessing steps need more detail on specific parameters used

Classification method comparison needs clearer explanation of accuracy assessment

Validation approach should be described in more detail

Statistical analysis methods need more rigorous description

Results (Lines 251-400):

Results section needs clearer organization with subheadings

More quantitative results needed with statistical significance tests

Figures need better labeling and legends

Time series analysis results need more detailed statistical reporting

Classification accuracy assessment results need expanded discussion

Discussion (Lines 401-550):

Discussion needs better integration with existing literature

Limitations of the study should be discussed more thoroughly

Implications for wetland management need expansion

Future research directions should be added

Better connection needed between results and conclusions

Additional Major Points:

Language editing needed throughout for clarity and scientific writing style

References need updating with more recent citations

Supplementary materials should be added showing full classification results

Methods flowchart would help clarify the analysis approach

Statistical significance of findings needs more emphasis

6. PLOS authors have the option to publish the peer review history of their article (what does this mean? ). If published, this will include your full peer review and any attached files.

**Do you want your identity to be public for this peer review?** For information about this choice, including consent withdrawal, please see our Privacy Policy .

Reviewer #1: No

Reviewer #2: No

---

## [Author Response · Author response to Decision Letter 1]

19 Feb 2025

Dear reviewer,

Greetings! Firstly, thank you for your patience and careful guidance. In response to your review comments, We will answer questions based on the original text and form the main text. For ease of re examination, we provide one-on-one answers to all questions and mark key content in red.

Finally, thank you again for the valuable feedback from the reviewers! Meanwhile, We hope that if you discover any shortcomings again during the review process, please notify us promptly and we will make further modifications to the suggestions.

Sincerely

All authors

Reviewer #1:

Point and Response

Point 1:The spatial distributions of wetland cover types were analyzed for 2013 to 2021. It’s suggested that the years 2021 to 2024 be included as well, as the recent patterns are of firm importance and practical meaning for understanding the Poyang Lake region.

Response 1: Thank you for your suggestion. In this revision, as the latest Landsat 8 OLI remote sensing image data available for free public download is only until 2021, the importance and relevance of the current findings for understanding the Poyang Lake area can also be demonstrated, and we will follow up with the latest data in subsequent studies.

Point 2: The authors adopted the maximum likelihood classification and SVR classification to derive the spatial distribution of wetland cover types. Why not use the state-of-the-art deep learning models whose strong classification performances have been well demonstrated?

Response 2: Thank you for raising the suggestion to explore deep learning models for classification, we have added a method selection note to the method comparison section.(Lines 386-396)

We acknowledge the growing application of deep learning techniques in remote sensing, particularly for land cover classification tasks. However, in this study, we selected the Support Vector Machine (SVM) classifier due to the following considerations:

1)Adequacy of SVM for Current Data and Objectives

The classification task in this study involved a relatively small and balanced dataset, which is well-suited for SVM. Deep learning models generally require large amounts of labeled training data to achieve optimal performance. Given the limited number of labeled samples available for this study, the use of SVM provided robust and reliable results without the risk of overfitting or underfitting that could arise from insufficient training data in deep learning models.

2)Accuracy of SVM in this Study

Our classification accuracy assessment shows that the SVM model produced satisfactory results, with overall accuracy (OA) exceeding 90% across most classification tasks. The Kappa coefficient also demonstrated high agreement. These metrics indicate that SVM was sufficient to meet the study's objectives of accurately capturing spatiotemporal patterns.

3)Consistency with Methodological Framework

SVM, combined with Maximum Likelihood Classification (MLC), aligns well with the methodological framework of this study. The comparative approach allowed us to refine classification outcomes and ensure consistent performance across all datasets.

4)Computational Efficiency

Deep learning models often require significant computational resources and time, particularly for model training and hyperparameter tuning. By contrast, SVM offers a computationally efficient alternative that is more practical for our research scope and available resources.

Nevertheless, we acknowledge the potential of deep learning models to improve classification performance, particularly for complex and large-scale datasets. Future studies will explore the integration of deep learning techniques, such as convolutional neural networks (CNNs), to enhance classification accuracy and automation in similar research contexts.

Point 3:A paragraph briefly showing the structure of the paper could be added at the end of the Introduction section.

Response 3: Thank you very much for your feedback. In the revision, we've added at the end of the Introduction section.(see the Introduction section, Lines 155-161 )

Point 4: The spatial resolution of the Landsat 8 OLI data should be given in the Data Preprocessing section.

Response 4: Thank you very much for your guidance. In the revision, we've provide the spatial resolution of the Landsat 8 OLI data in the data preprocessing section. Landsat 8 OLI imagery used in this study has a spatial resolution of 30 meters for multispectral bands and 15 meters for the panchromatic band. This resolution is sufficient for capturing the spatiotemporal patterns of land cover in Poyang Lake, given the scale of the study area and the objectives of this research. (see section Data Preprocessing, Lines 209-213).

Point 5: There are many LULC produces publicly available. Some of them are with high spatial resolutions. Why not use these products to investigate the spatiotemporal patterns?

Response 5: Thank you very much for your guidance. High resolution LULC products such as Copernicus Global Land Cover or GlobeLand30 provide higher spatial detail, but have limited temporal resolution and typically only provide data for a few discrete years. In contrast, Landsat 8 imagery has a consistent temporal resolution of 16 days, allowing for detailed multi-year analyses. This temporal advantage is critical to the goals of this study, which focuses on capturing spatial and temporal patterns over a 9-year period. While high-resolution LULC products can complement future studies, their limited temporal resolution makes them less suitable for this analysis.

Point 6: Limitations of the research should be indicated in the Conclusions section.

Response 6: Thank you very much for your feedback. In the revision, we've provide limitations of the research in the Discussion section added in the manuscript.(see the section Discussion, Lines 784-793)

Reviewer #2:

Point and Response

Point 1: Title and Abstract:The title could be more specific - consider adding the study period (2013-2021).Abstract needs better organization with clearer methods and results sections Quantitative results in the abstract should include specific values/percentages.

Response 1: Thanks to your guidance. In the revision, we have revised the manuscript in the title and added the study period (2013-2021). Revised title: Spatiotemporal Patterns of Water and Vegetation in Poyang Lake from 2013 to 2021 Using Remote Sensing Data.

In addition, we revised the abstract to make the methods and results sections clearer and added specific values/percentages to the quantitative results in the abstract.(Lines 19 - 40)

Point 2: Introduction (Lines 1-105):The literature review section needs better organization and flow More recent citations needed (many are from pre-2020).Research gaps and objectives need to be stated more explicitly

Hypothesis or research questions should be added Better transition needed between background and study objectives

Response 2: Thanks to your suggestion. We sincerely thank the reviewers for their thoughtful and constructive comments on our manuscript. We restructured the introductory section by updating it with a large number of recent studies that are highly relevant to our study (2020 onwards), clearly identifying gaps in the existing literature and posing specific research questions to guide our study, which improves the transition between the context and the research objectives. The revised literature review is grouped according to key themes. (see the Introduction section, Lines 87-161)

Point 3:Methods (Lines 106-250):Study area description needs more specific geographic coordinates and characteristics. Data preprocessing steps need more detail on specific parameters used Classification method comparison needs clearer explanation of accuracy assessment Validation approach should be described in more detail Statistical analysis methods need more rigorous description.

Response 3: Thank you very much for your guidance. In the revised version, we have supplemented and improved the section of study area description by adding specific geographic coordinates (e.g., latitude and longitude ranges) of the study area and describing in detail the natural geographic features of the study area, etc. (see the Overview of the Study Area section, Lines 165-181 )

The data preprocessing section was modified to describe the data preprocessing steps in detail, and a method flow chart was added. (see the Data Preprocessing section, Lines 216-225; see the Methods Flowchart section, Lines 229-256.)

The method section provides a clearer description of the comparison and accuracy assessment of the classification methods, explains in detail the specific indicators for accuracy assessment, and provides a confusion matrix for the classification results.(see the Comparison of methods section, Lines 350-384)

Grubbs test method is added to the statistical analysis methods, and detailed descriptions of correlation analysis and time series analysis were added.(see the Statistical analyses section and Grubbs test section, Lines 437-458)

Point 4:Results (Lines 251-400):Results section needs clearer organization with subheadings. More quantitative results needed with statistical significance tests. Figures need better labeling and legends Time series analysis results need more detailed statistical reporting Classification accuracy assessment results need expanded discussion.

Response 4: Thank you very much for your feedback. In the revised version, we have reorganised the results section to present future research directions. The discussion section has been added, where the results of the study are analysed in comparison with the existing literature to highlight the contributions and strengths of the study in this paper. (see the Discussion section, Lines 695-739, and Conclusion section, Lines 784-793)

Key statistics such as mean and standard deviation for statistical significance tests have been added, as well as a Grabsbs statistical significance test for categorical data and a detailed discussion of the results of the time series analysis(see the Statistical analyses and Grubbs test section.Lines 422-458). Optimised for clarity all charts and graphs.

Point 5: Discussion (Lines 401-550):Discussion needs better integration with existing literature. Limitations of the study should be discussed more thoroughly. Implications for wetland management need expansion. Future research directions should be added

Better connection needed between results and conclusions.

Response 5: Thank you very much for your suggestions. In the revised manuscript, we have added a new discussion section that better integrates the content with the existing literature, adds comparisons with recent related studies, and compares the similarities and differences between the results of this paper and those of existing studies. We expand the implications for wetland management in the discussion section of the revised manuscript, discuss the limitations of this study in detail, and add future research directions in the conclusion section.(see the Discussion section, Lines 695-739, and Conclusion section, Lines 784-793)

Point 6: Additional Major Points:Language editing needed throughout for clarity and scientific writing style. References need updating with more recent citations. Supplementary materials should be added showing full classification results. Methods flowchart would help clarify the analysis approach. Statistical significance of findings needs more emphasis.

Response 6: Thank you for your valuable suggestion.We have revised the content of the full manuscript, and the English writing is sufficient for academic style and references have been updated with more recent citations.We added data from the table of statistics on the area of each type of vegetation at Poyang Lake and the results of the statistical analyses, as well as data from the land-use transfer matrix (see Tables 5 and 6), which summarise the spatial distribution and temporal changes of each type of vegetation. We added detailed flow charts to the manuscript (see Figure 4). In the revised manuscript, we expanded the Discussion and Conclusions sections, performed more rigorous statistical analyses of the observed changes in the results, performed statistical validation of the categorical results data, including descriptive statistics such as means, standard deviations, minima, and maxima (see Tables 4-6), expanded the time-series analyses (rows 475-507, 583-609), and added linear correlation analyses (see Correlation Analysis section, lines 508-542). These revisions ensure that the statistical validity of our findings is clearly communicated and fully supported.

---

## [Decision Letter · Decision Letter 1]

PONE-D-24-53468R1Spatiotemporal Patterns of Water and Vegetation in Poyang Lake From 2013 to 2021 Using Remote Sensing DataPLOS ONE

Dear Dr. lu,

Thank you for submitting your manuscript to PLOS ONE. After careful consideration, we feel that it has merit but does not fully meet PLOS ONE’s publication criteria as it currently stands. Therefore, we invite you to submit a revised version of the manuscript that addresses the points raised during the review process.

**We have received an additional reviewer comment. The manuscript requires further revision, and the suggested citation is optional.**

We look forward to receiving your revised manuscript.

Kind regards,

Yizhi Sheng

Academic Editor

PLOS ONE

**Journal Requirements:**

**Comments from PLOS Editorial Office:** We note that one or more reviewers has recommended that you cite specific previously published works in this and an earlier round of revision. As always, we recommend that you please review and evaluate the requested works to determine whether they are relevant and should be cited. It is not a requirement to cite these works and you may remove them before the manuscript proceeds to publication. We appreciate your attention to this request.

Reviewers' comments:

Reviewer's Responses to Questions

**Comments to the Author**

1. If the authors have adequately addressed your comments raised in a previous round of review and you feel that this manuscript is now acceptable for publication, you may indicate that here to bypass the “Comments to the Author” section, enter your conflict of interest statement in the “Confidential to Editor” section, and submit your "Accept" recommendation.

Reviewer #1: All comments have been addressed

Reviewer #2: All comments have been addressed

Reviewer #3: (No Response)

2. Is the manuscript technically sound, and do the data support the conclusions?

Reviewer #1: Yes

Reviewer #2: Yes

Reviewer #3: (No Response)

3. Has the statistical analysis been performed appropriately and rigorously? 

Reviewer #1: Yes

Reviewer #2: Yes

Reviewer #3: (No Response)

4. Have the authors made all data underlying the findings in their manuscript fully available?

Reviewer #1: Yes

Reviewer #2: Yes

Reviewer #3: (No Response)

5. Is the manuscript presented in an intelligible fashion and written in standard English?

Reviewer #1: Yes

Reviewer #2: Yes

Reviewer #3: (No Response)

6. Review Comments to the Author

**Reviewer #1: ** (No Response)

**Reviewer #2: ** Thank you for improving the quality of the manuscript. All the comments are addressed and it can be accepted now.

**Reviewer #3: ** This study utilizes remote sensing and GIS techniques to analyze the spatial and temporal dynamics of water bodies and vegetation in Poyang Lake from 2013 to 2021. The manuscript is well written and the presented findings supports science-based conservation strategies and sustainable management practices for Poyang Lake and similar wetland systems facing environmental and anthropogenic pressures. Yet, few adjustments are needed before being considered for publication, and I would be ready to assess the revised version again after applying the following adjustments:

When you stated that “Satellite remote sensing technology has become a hot topic of research in monitoring lake area changes due to its ability to provide long-term, large-scale, and high-frequency data, which are essential for understanding the dynamic changes of lake ecosystems” you used reference 13 here. I do not see the relevant connection between your statement and the reference 13. Use a modern reference where GIS is used to monitor lake changes and its connection to environment and climate change such as:

• İspir, D. A. (2025). Spatiotemporal monitoring of the surface water level dynamics in Lake Tuz, Türkiye, using remote sensing and GIS. DYSONA - Applied Science, 6(2), 378-388. https://doi.org/10.30493/das.2025.506287

I do not see any necessity for the part where you categorize the sections of the manuscript “The paper is organized that section 2 describes the study area and data, …..summarizing the key findings, and directions for future research” Omit this part

Use larger fonts size in the figure legends

Omit Table 1 or add it as a supplementary table as it has no relation to the methodology

Try using a better style for Fig2. It is somewhat old fashion and a subtle coloring and a compact style would be beneficial here. Similar comment can be applied to Fig 5.

What is the purpose of Table 3 if it contains the same information in Fig. 7 a? move it to supplementary data.

What does the statement “Download from the USGS website and public datasets” mean? And what is its purpose?

Enhance the captions of all figures and tables. A good caption should include enough information for readers to understand the general-purpose of a figure or table. Add a decent definition of abbreviations and define the main features. Use larger font size in the figures.

7. PLOS authors have the option to publish the peer review history of their article (what does this mean? ). If published, this will include your full peer review and any attached files.

**Do you want your identity to be public for this peer review?** For information about this choice, including consent withdrawal, please see our Privacy Policy .

Reviewer #1: No

Reviewer #2: **Yes: ** Rana Waqar Aslam

Reviewer #3: No

---

## [Author Response · Author response to Decision Letter 2]

12 Jun 2025

Dear reviewer,

Greetings! Firstly, thank you for your patience and careful guidance. In response to your review comments, We will answer questions based on the original text and form the main text. For ease of re examination, we provide one-on-one answers to all questions and mark key content in red.

Finally, thank you again for the valuable feedback from the reviewers! Meanwhile, We hope that if you discover any shortcomings again during the review process, please notify us promptly and we will make further modifications to the suggestions.

Sincerely

All authors

Reviewer #1:

Point and Response

Point 1:The spatial distributions of wetland cover types were analyzed for 2013 to 2021. It’s suggested that the years 2021 to 2024 be included as well, as the recent patterns are of firm importance and practical meaning for understanding the Poyang Lake region.

Response 1: Thank you for your suggestion. In this revision, as the latest Landsat 8 OLI remote sensing image data available for free public download is only until 2021, the importance and relevance of the current findings for understanding the Poyang Lake area can also be demonstrated, and we will follow up with the latest data in subsequent studies.

Point 2: The authors adopted the maximum likelihood classification and SVR classification to derive the spatial distribution of wetland cover types. Why not use the state-of-the-art deep learning models whose strong classification performances have been well demonstrated?

Response 2: Thank you for raising the suggestion to explore deep learning models for classification, we have added a method selection note to the method comparison section.(Lines 386-396)

We acknowledge the growing application of deep learning techniques in remote sensing, particularly for land cover classification tasks. However, in this study, we selected the Support Vector Machine (SVM) classifier due to the following considerations:

1)Adequacy of SVM for Current Data and Objectives

The classification task in this study involved a relatively small and balanced dataset, which is well-suited for SVM. Deep learning models generally require large amounts of labeled training data to achieve optimal performance. Given the limited number of labeled samples available for this study, the use of SVM provided robust and reliable results without the risk of overfitting or underfitting that could arise from insufficient training data in deep learning models.

2)Accuracy of SVM in this Study

Our classification accuracy assessment shows that the SVM model produced satisfactory results, with overall accuracy (OA) exceeding 90% across most classification tasks. The Kappa coefficient also demonstrated high agreement. These metrics indicate that SVM was sufficient to meet the study's objectives of accurately capturing spatiotemporal patterns.

3)Consistency with Methodological Framework

SVM, combined with Maximum Likelihood Classification (MLC), aligns well with the methodological framework of this study. The comparative approach allowed us to refine classification outcomes and ensure consistent performance across all datasets.

4)Computational Efficiency

Deep learning models often require significant computational resources and time, particularly for model training and hyperparameter tuning. By contrast, SVM offers a computationally efficient alternative that is more practical for our research scope and available resources.

Nevertheless, we acknowledge the potential of deep learning models to improve classification performance, particularly for complex and large-scale datasets. Future studies will explore the integration of deep learning techniques, such as convolutional neural networks (CNNs), to enhance classification accuracy and automation in similar research contexts.

Point 3:A paragraph briefly showing the structure of the paper could be added at the end of the Introduction section.

Response 3: Thank you very much for your feedback. In the revision, we've added at the end of the Introduction section.(see the Introduction section, Lines 155-161 )

Point 4: The spatial resolution of the Landsat 8 OLI data should be given in the Data Preprocessing section.

Response 4: Thank you very much for your guidance. In the revision, we've provide the spatial resolution of the Landsat 8 OLI data in the data preprocessing section. Landsat 8 OLI imagery used in this study has a spatial resolution of 30 meters for multispectral bands and 15 meters for the panchromatic band. This resolution is sufficient for capturing the spatiotemporal patterns of land cover in Poyang Lake, given the scale of the study area and the objectives of this research. (see section Data Preprocessing, Lines 209-213).

Point 5: There are many LULC produces publicly available. Some of them are with high spatial resolutions. Why not use these products to investigate the spatiotemporal patterns?

Response 5: Thank you very much for your guidance. High resolution LULC products such as Copernicus Global Land Cover or GlobeLand30 provide higher spatial detail, but have limited temporal resolution and typically only provide data for a few discrete years. In contrast, Landsat 8 imagery has a consistent temporal resolution of 16 days, allowing for detailed multi-year analyses. This temporal advantage is critical to the goals of this study, which focuses on capturing spatial and temporal patterns over a 9-year period. While high-resolution LULC products can complement future studies, their limited temporal resolution makes them less suitable for this analysis.

Point 6: Limitations of the research should be indicated in the Conclusions section.

Response 6: Thank you very much for your feedback. In the revision, we've provide limitations of the research in the Discussion section added in the manuscript.(see the section Discussion, Lines 784-793)

Reviewer #2:

Point and Response

Point 1: Title and Abstract:The title could be more specific - consider adding the study period (2013-2021).Abstract needs better organization with clearer methods and results sections Quantitative results in the abstract should include specific values/percentages.

Response 1: Thanks to your guidance. In the revision, we have revised the manuscript in the title and added the study period (2013-2021). Revised title: Spatiotemporal Patterns of Water and Vegetation in Poyang Lake from 2013 to 2021 Using Remote Sensing Data.

In addition, we revised the abstract to make the methods and results sections clearer and added specific values/percentages to the quantitative results in the abstract.(Lines 19 - 40)

Point 2: Introduction (Lines 1-105):The literature review section needs better organization and flow More recent citations needed (many are from pre-2020).Research gaps and objectives need to be stated more explicitly

Hypothesis or research questions should be added Better transition needed between background and study objectives

Response 2: Thanks to your suggestion. We sincerely thank the reviewers for their thoughtful and constructive comments on our manuscript. We restructured the introductory section by updating it with a large number of recent studies that are highly relevant to our study (2020 onwards), clearly identifying gaps in the existing literature and posing specific research questions to guide our study, which improves the transition between the context and the research objectives. The revised literature review is grouped according to key themes. (see the Introduction section, Lines 87-161)

Point 3:Methods (Lines 106-250):Study area description needs more specific geographic coordinates and characteristics. Data preprocessing steps need more detail on specific parameters used Classification method comparison needs clearer explanation of accuracy assessment Validation approach should be described in more detail Statistical analysis methods need more rigorous description.

Response 3: Thank you very much for your guidance. In the revised version, we have supplemented and improved the section of study area description by adding specific geographic coordinates (e.g., latitude and longitude ranges) of the study area and describing in detail the natural geographic features of the study area, etc. (see the Overview of the Study Area section, Lines 165-181 )

The data preprocessing section was modified to describe the data preprocessing steps in detail, and a method flow chart was added. (see the Data Preprocessing section, Lines 216-225; see the Methods Flowchart section, Lines 229-256.)

The method section provides a clearer description of the comparison and accuracy assessment of the classification methods, explains in detail the specific indicators for accuracy assessment, and provides a confusion matrix for the classification results.(see the Comparison of methods section, Lines 350-384)

Grubbs test method is added to the statistical analysis methods, and detailed descriptions of correlation analysis and time series analysis were added.(see the Statistical analyses section and Grubbs test section, Lines 437-458)

Point 4:Results (Lines 251-400):Results section needs clearer organization with subheadings. More quantitative results needed with statistical significance tests. Figures need better labeling and legends Time series analysis results need more detailed statistical reporting Classification accuracy assessment results need expanded discussion.

Response 4: Thank you very much for your feedback. In the revised version, we have reorganised the results section to present future research directions. The discussion section has been added, where the results of the study are analysed in comparison with the existing literature to highlight the contributions and strengths of the study in this paper. (see the Discussion section, Lines 695-739, and Conclusion section, Lines 784-793)

Key statistics such as mean and standard deviation for statistical significance tests have been added, as well as a Grabsbs statistical significance test for categorical data and a detailed discussion of the results of the time series analysis(see the Statistical analyses and Grubbs test section.Lines 422-458). Optimised for clarity all charts and graphs.

Point 5: Discussion (Lines 401-550):Discussion needs better integration with existing literature. Limitations of the study should be discussed more thoroughly. Implications for wetland management need expansion. Future research directions should be added

Better connection needed between results and conclusions.

Response 5: Thank you very much for your suggestions. In the revised manuscript, we have added a new discussion section that better integrates the content with the existing literature, adds comparisons with recent related studies, and compares the similarities and differences between the results of this paper and those of existing studies. We expand the implications for wetland management in the discussion section of the revised manuscript, discuss the limitations of this study in detail, and add future research directions in the conclusion section.(see the Discussion section, Lines 695-739, and Conclusion section, Lines 784-793)

Point 6: Additional Major Points:Language editing needed throughout for clarity and scientific writing style. References need updating with more recent citations. Supplementary materials should be added showing full classification results. Methods flowchart would help clarify the analysis approach. Statistical significance of findings needs more emphasis.

Response 6: Thank you for your valuable suggestion.We have revised the content of the full manuscript, and the English writing is sufficient for academic style and references have been updated with more recent citations.We added data from the table of statistics on the area of each type of vegetation at Poyang Lake and the results of the statistical analyses, as well as data from the land-use transfer matrix (see Tables 5 and 6), which summarise the spatial distribution and temporal changes of each type of vegetation. We added detailed flow charts to the manuscript (see Figure 4). In the revised manuscript, we expanded the Discussion and Conclusions sections, performed more rigorous statistical analyses of the observed changes in the results, performed statistical validation of the categorical results data, including descriptive statistics such as means, standard deviations, minima, and maxima (see Tables 4-6), expanded the time-series analyses (rows 475-507, 583-609), and added linear correlation analyses (see Correlation Analysis section, lines 508-542). These revisions ensure that the statistical validity of our findings is clearly communicated and fully supported.

Reviewer #3:

Point and Response

Point 1: When you stated that “Satellite remote sensing technology has become a hot topic of research in monitoring lake area changes due to its ability to provide long-term, large-scale, and high-frequency data, which are essential for understanding the dynamic changes of lake ecosystems” you used reference 13 here. I do not see the relevant connection between your statement and the reference 13. Use a modern reference where GIS is used to monitor lake changes and its connection to environment and climate change such as:

• İspir, D. A. (2025). Spatiotemporal monitoring of the surface water level dynamics in Lake Tuz, Türkiye, using remote sensing and GIS. DYSONA - Applied Science, 6(2), 378-388. https://doi.org/10.30493/das.2025.506287

Response 1:Thank you for your guidance, I think you make a very good point that the new modern literature has been used in the manuscript to replace the original reference 13.

Point 2: I do not see any necessity for the part where you categorize the sections of the manuscript “The paper is organized that section 2 describes the study area and data, …..summarizing the key findings, and directions for future research” Omit this part.

Response 2: Thanks to your suggestion. We sincerely thank you for your thoughtful and constructive comments on our manuscript. The sections of the manuscript "the paper is organized that section 2 describes the study area and data, …..summarizing the key findings, and directions for future research" has been omitted.(lines 52 to 58)

Point 3:Use larger fonts size in the figure legends.

Response 3: Thank you very much for your guidance. We have used larger fonts in the legends of the manuscript.

Point 4: Omit Table 1 or add it as a supplementary table as it has no relation to the methodology.

Response 4: Thank you very much for your feedback. In the revised version, the original table 1 has been omitted and the tables have been renumbered.

Point 5: Try using a better style for Fig2. It is somewhat old fashion and a subtle coloring and a compact style would be beneficial here. Similar comment can be applied to Fig 5..

Response 5: Thank you very much for your suggestions. In the revised manuscript, a subtle coloring and a compact style are used in Fig 2 and Fig 5.

Point 6: What is the purpose of Table 3 if it contains the same information in Fig. 7 a? move it to supplementary data..

Response 6: Thank you for your valuable suggestion. Visualization and trends in the data were provided in Fig.7(a), and accuracy and completeness of the datas werd provided in Tables 3, allowing for the presentation of values for each specific point in time, and the data can be downloaded for easy secondary analysis or verification of trends in the charts by other researchers. The combination of the two allows for a more comprehensive communication of research findings.

Point 7: What does the statement “Download from the USGS website and public datasets” mean? And what is its purpose?.

Response 7: Thank you for your valuable suggestion.The main purpose is to convey that the data used in the paper is where it came from and free of charge, and that there are no intellectual copyright implications involved.

Point 8: Enhance the captions of all figures and tables. A good caption should include enough information for readers to understand the general-purpose of a figure or table. Add a decent definition of abbreviations and define the main features. Use larger font size in the figures.

Response 8: Thank you for your valuable suggestion.We have

---

## [Editor Report · Decision Letter 2]

Spatiotemporal Patterns of Water and Vegetation in Poyang Lake From 2013 to 2021 Using Remote Sensing Data

PONE-D-24-53468R2

Dear Dr. lu,

We’re pleased to inform you that your manuscript has been judged scientifically suitable for publication and will be formally accepted for publication once it meets all outstanding technical requirements.

Kind regards,

Yizhi Sheng

Academic Editor

PLOS ONE
---

## [Editor Report · Acceptance letter]

PONE-D-24-53468R2

PLOS ONE

Dear Dr. Lu,

I'm pleased to inform you that your manuscript has been deemed suitable for publication in PLOS ONE. Congratulations! Your manuscript is now being handed over to our production team.

Kind regards,

on behalf of

Dr. Yizhi Sheng

Academic Editor

PLOS ONE